# Long-term consistency in chimpanzee consolation behaviour reflects empathetic personalities

Christine E. Webb[1,2], Teresa Romero[1,3,4], Becca Franks[5] & Frans B. M. de Waal[1,2]

In contrast to a wealth of human studies, little is known about the ontogeny and consistency of empathy-related capacities in other species. Consolation—post-conflict affiliation from uninvolved bystanders to distressed others—is a suggested marker of empathetic concern in non-human animals. Using longitudinal data comprising nearly a decade of observations on over 3000 conflict interactions in 44 chimpanzees (*Pan troglodytes*), we provide evidence for relatively stable individual differences in consolation behaviour. Across development, individuals consistently differ from one another in this trait, with higher consolatory tendencies predicting better social integration, a sign of social competence. Further, similar to recent results in other ape species, but in contrast to many human self-reported findings, older chimpanzees are less likely to console than are younger individuals. Overall, given the link between consolation and empathy, these findings help elucidate the development of individual socio-cognitive and -emotional abilities in one of our closest relatives.

[1] Living Links, Yerkes National Primate Research Center and Psychology Department, Emory University, Atlanta, GA 30322, USA. [2] Utrecht University, Utrecht 3508 TC, The Netherlands. [3] School of Life Sciences, University of Lincoln, Lincoln LN6 7TS, UK. [4] Department of Cognitive and Behavioural Science, University of Tokyo, Tokyo 153-0041, Japan. [5] Animal Welfare Program, University of British Columbia, Vancouver, BC, Canada V6T 1Z4. Correspondence and requests for materials should be addressed to C.E.W. (email: cwebb218@gmail.com) or to T.R. (email: tromero@lincoln.ac.uk)

Empathy, the ability to share and understand the emotions and cognitions of others[1–3], is a core component of social development. Not only does empathy enable individuals to coordinate inner states and cooperate towards joint goals, it allows partners to establish and maintain successful relationships. Accumulating evidence from research in humans and non-human animals (hereafter, animals) reveals that different facets of empathy may be present through much of the evolutionary history of vertebrates[3, 4]—with basic forms, including emotional contagion, being found in most extant social species[5, 6]. Animal research has further revealed that similarity, familiarity, and social closeness facilitate the expression of empathy, which also applies to human empathetic processes[2–4, 7, 8].

The human developmental literature indicates that empathy-related responses emerge early in life. Signs of concern for and assistance to distressed others are already present at 2 years of age[9], with more recent reviews suggesting that empathetic concern arises during infancy[10]. Moreover, empathy is generally thought to increase in both frequency and complexity over the lifespan[11]. This shift involves changes in spontaneous empathetic responses towards others, prosocial behaviours to reduce others' distress, and cognitive perspective-taking abilities[12].

Human longitudinal studies support the notion that there are stable individual differences in empathetic responding[12–16]. Considering the developmental trends described above, it is worth noting that such differences imply rank-order or relative stability rather than behavioural tendencies that do not change, i.e., absolute stability. This relative individual consistency suggests that empathy may be conceptualized as part of a broader pro-social personality domain that develops early in life and impacts various aspects of an individual's sociality later in life, such as its social competence[17]. However, while these studies provide rather convincing evidence that individual variation in empathy is relatively stable over the course of human development, most have emphasized consistency within, rather than across, developmental stages (see ref. [15] for an exception).

Many animals are also able to recognize others' emotions and respond accordingly. The best-documented example of empathetic concern in another species is consolation behaviour, i.e., spontaneous affiliation directed by an uninvolved bystander to a recent recipient of aggression[18]. This definition excludes other types of post-conflict third-party affiliation, such as third-party contacts sought by the conflict participants themselves or made with the aggressor. While the precise cognitive and emotional capacities required for consolation remain hard to elucidate, there are behavioural indicators that it alleviates the recipient's distress[19–21], occurs most often amongst close social partners[22, 23], and follows other predictions derived from an empathy-based hypothesis[3, 4, 24]. Alternative functions of consolation have been proposed[25], such as that it serves as a form of mediated reconciliation or as a mechanism to protect the bystander from redirected aggression. Although these alternatives should not be readily dismissed, the most recent review of the evidence to date supports the empathy hypothesis—not just in primates, but across diverse mammalian species[26]. For example, a recent rodent study illuminated underlying neural mechanisms—not only do consolers match the fear response, anxiety-like behaviours, and corticosterone increase of the stressed recipient, but consolation appears to be oxytocin-dependent[27] (see also ref. [28]). For these reasons, animal consolation behaviour is often considered homologous in both form and function to human empathy-related responding[3, 26].

Research in non-human primates shows that individuals of all age-classes provide reassuring contact to distressed conspecifics, and that consolation tendency differs across age groups[20, 23, 29]. A set of recent studies found that juvenile bonobo bystanders offered consolation significantly more than did adolescents or adults[20, 23]. Levels of consolation also appear to be highest among infant and juvenile lowland gorillas as compared to older group members[29]. However, these studies failed to examine if the reported differences reflect age characteristics or stable individual differences, or both. More broadly, insufficient sample sizes and limited longitudinal data have precluded formal conclusions regarding developmental questions. Perhaps for these reasons, to date the vast majority of studies on animal consolation have excluded immature individuals or neglected to explicitly explore age as a factor in their analyses. Further, prior research has largely emphasized relational determinants of post-conflict behaviour over the potential role of consistent individual variation. Thus, despite the growing interest in the evolutionary roots of human empathy and altruism, it remains unknown whether stable individual differences in consolation and other putative behavioural manifestations of empathy (see ref. [26]) also exist in other animals.

Empathy is commonly associated with the Agreeableness domain described by widely used psychometric personality models[30]. Those high in Agreeableness are perceived as being sympathetic, sensitive and helpful towards others[30–35]. In humans, longitudinal work has revealed that facets of Agreeableness[31] along with similar traits (reviewed in ref. [32]) are relatively consistent, with many studies pointing to age-related increases. There is also evidence for the stability of Agreeableness in chimpanzees[33, 34], and for its expression to be higher among older individuals[35]. However, unlike the approach taken in the current work, these studies were based on questionnaire ratings (e.g., wherein observers code for personality descriptor adjectives like those listed above) rather than a behavioural measure.

The present study used a long-term data set of chimpanzee (Pan troglodytes) conflict and post-conflict interactions to investigate the stability of individual differences in consolation behaviour. First, we predicted that individual variation in consolation tendency would be present after controlling for numerous variables previously shown to influence the occurrence of this behaviour (e.g., the nature of the relationship between bystander and recipient). Additionally, similar to human empathetic behaviour[15], we expected that an individual's tendency to console would be relatively consistent across its lifespan. Finally, and also in line with human research findings[17], we predicted that individuals with stronger tendencies to console would exhibit higher social integration, a measure of social competence. Nearly a decade of observations on a large number of subjects of all age-classes has yielded data on over 3000 spontaneously occurring agonistic conflicts, providing a unique opportunity to test these predictions. Not only this, it allowed us to examine whether chimpanzee consolation increased (as would be expected from human research) or decreased (as has been found in other apes) over the course of development. Importantly, the longitudinal approach taken by the current research analyzed consolation at all developmental stages (infancy to adulthood) and thus afforded novel insights to the individual stability and ontogenetic trajectory of this presumed empathy-driven behaviour.

Our results demonstrate that individual differences in chimpanzee consolation behaviour are relatively stable across development. Beyond its individual repeatability, consolation generally declines over the lifespan, with older chimpanzees being less likely to console than younger chimpanzees. We also find support for a relation between consolation and social competence, such that high consolers are more socially integrated than low consolers. Given that consolation is considered a marker of empathy in human and non-human animals, its expression and trajectory in one of our closest primate relatives can provide key insights to the evolution of other-oriented responses that are fundamental to social life.

**Table 1 Results of GLMM testing for individual differences in consolation**

| Variable | | b | SE | CI$_{95}$ | Test statistic | P-value |
|---|---|---|---|---|---|---|
| *Fixed effects* | | | | | Z | |
| Consolation opportunities | | 0.013 | 0.004 | 0.006 to 0.021 | 3.47 | **0.001** |
| Bystander affiliation rate | | 0.089 | 0.102 | −0.111 to 0.288 | 0.87 | 0.386 |
| Dyad affiliation level | | 0.780 | 0.152 | 0.482 to 1.077 | 5.14 | **0.000** |
| Dyad kinship | | 1.220 | 0.195 | 0.838 to 1.602 | 6.25 | **0.000** |
| Bystander rank | | | | | | |
|   Low | | −1.232 | 0.316 | −1.852 to −0.613 | −3.90 | **0.000** |
|   High | | 0.121 | 0.424 | −0.710 to 0.951 | 0.28 | 0.776 |
| Bystander sex | | −0.117 | 0.237 | −0.582 to 0.348 | −0.49 | 0.623 |
| Bystander rank*sex | | −0.431 | 0.517 | −1.444 to 0.582 | −0.83 | 0.404 |
| Bystander age | | | | | | |
|   Juvenile | | −0.478 | 0.274 | −1.015 to 0.059 | −1.74 | 0.081 |
|   Adolescent | | −1.041 | 0.382 | −1.789 to −0.293 | −2.73 | **0.006** |
|   Adult | | −1.251 | 0.399 | −2.034 to −0.469 | −3.13 | **0.002** |
| Recipient rank | | | | | | |
|   Low | | −0.115 | 0.313 | −0.728 to 0.498 | −0.37 | 0.712 |
|   High | | 0.052 | 0.235 | −0.409 to 0.514 | 0.22 | 0.824 |
| Recipient sex | | −0.263 | 0.182 | −0.620 to 0.093 | −1.45 | 0.148 |
| Recipient age | | | | | | |
|   Juvenile | | −0.119 | 0.257 | −0.623 to 0.386 | −0.46 | 0.645 |
|   Adolescent | | 0.090 | 0.382 | −0.659 to 0.839 | 0.24 | 0.813 |
|   Adult | | 0.247 | 0.391 | −0.507 to 1.030 | 0.63 | 0.527 |
| *Random effect* | | | | | $\chi^2$ | |
| Bystander | Full model | 0.433 | 0.097 | 0.279 to 0.670 | 17.54 | **0.000** |
| Recipient | | 0.270 | 0.097 | 0.134 to 0.545 | | |
| Recipient | Partial model | 0.269 | 0.093 | 0.136 to 0.531 | 3.70 | **0.027** |

Significant P values <0.05 are shown in bold. Fixed effects are shown for the full model.

## Results

**Individual differences in consolation**. As Table 1 shows, we found evidence for consistent individual variation in bystander consolation while controlling for other factors shown by previous research to affect this behaviour, including the number of opportunities the bystander had to offer the recipient consolation which, unsurprisingly, was a significant predictor of the behaviour (generalized linear multilevel model (GLMM): $z = 3.47$, $b = 0.01$, $P = 0.001$). Our base GLMM revealed an effect of recipient (likelihood ratio test: $\chi^2(1) = 3.70$, $P = 0.027$), but including bystander as a random effect significantly improved model fit, revealing a bystander effect (full GLMM with both recipient and bystander as crossed-random effects vs. restricted GLMM with recipient as the only random effect (likelihood ratio test: $\chi^2(1) = 13.84$, $P < 0.001$)). Importantly, these results are reported accounting for the bystander's baseline affiliation rate (which did not significantly predict consolation's occurrence: $z = 0.87$, $b = 0.09$, $P = 0.39$), ruling out the possibility that individual differences in consolation are merely an artifact of the general tendency to affiliate.

The affiliation level between bystander-recipient dyads positively predicted the occurrence of consolation ($z = 5.14$, $b = 0.78$, $P < 0.001$) as did kinship ($z = 6.25$, $b = 1.22$, $P < 0.001$), confirming previous findings that consolation occurs most often in close relationships, including in these groups[22, 24]. Low-ranking bystanders were significantly less likely to provide consolation than medium-ranking bystanders ($z = -3.90$, $b = -1.23$, $P < 0.001$), but no other differences were found regarding bystander dominance.

As shown in Fig. 1, bystander age-class was also a significant predictor of consolation. Infants were significantly more likely to provide consolation than either adolescents or adults (infants compared to adolescents: $z = -2.73$, $b = -1.04$, $P = 0.006$; adults: $z = -3.13$, $b = -1.25$, $P = 0.002$), but not juveniles ($z = -1.74$, $b = -0.48$, $P = 0.081$). Planned contrasts confirmed these patterns

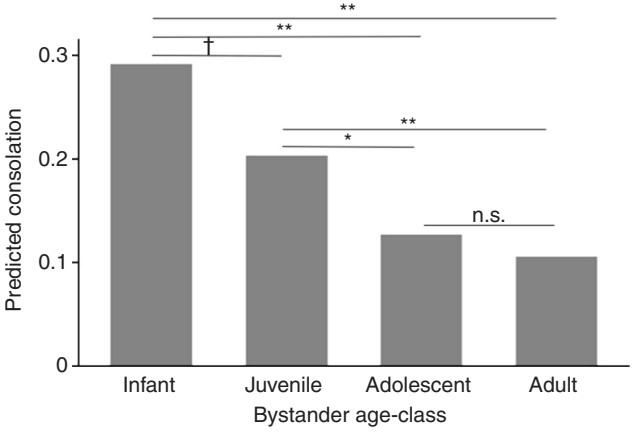

**Fig. 1** Predicted probability of consolation by bystander age-class. Values were calculated separately for each bystander age-class using the full GLMM (Table 1), with planned comparisons revealing whether differences among the age-classes were significant. **$P < 0.01$; *$P < 0.05$; †$P < 0.10$; n.s. = not significant

and further revealed that consolation was higher in juveniles than in either adolescents or adults (juvenile compared to adolescents: $z = 2.24$, $b = 0.56$, $P = 0.025$; adults: $z = 2.87$, $b = 0.77$, $P = 0.004$), but did not differ significantly between adolescents and adults ($z = 0.82$, $b = 0.21$, $P = 0.414$). Unlike bystanders, neither recipients' age-class nor rank were significant predictors of consolation (Table 1).

**Repeatability of consolation over the lifespan**. We then tested the relationship between consolation tendency from the youngest and oldest age period(s) on record for each individual. As Fig. 2

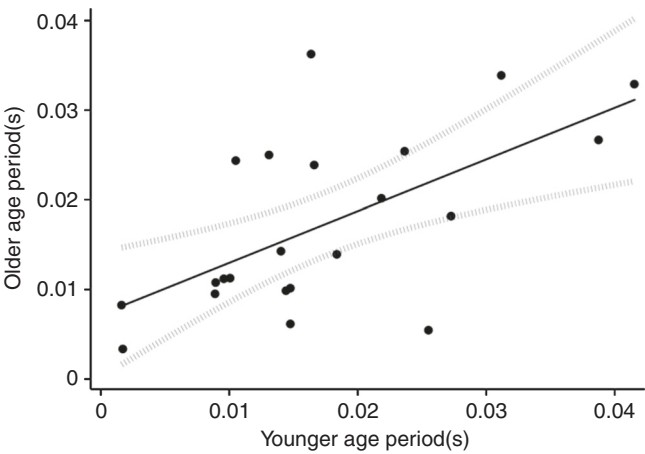

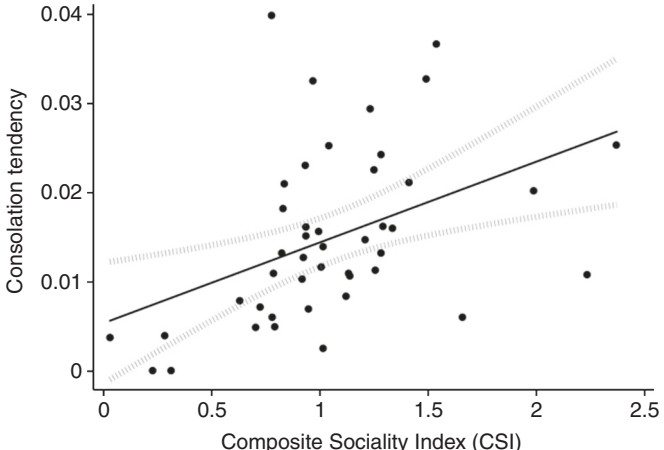

**Fig. 2** Consolation tendency is consistent across lifespan. Consolation tendencies for the youngest and oldest age-classes on record (respectively) were calculated by dividing a subject's total number of consolations by its total number of opportunities to console during each age period. We then compared consolation tendency from the youngest age period(s) on record to that of the oldest age period(s) on record for each subject. Only subjects who progressed through multiple age-classes over the observation period were considered ($N = 22$)

**Fig. 3** Consolation tendency and social competence by subject. A Composite Sociality Index (CSI) was generated for each subject ($N = 44$) from grooming and proximity scores, and compared to the tendency to console, calculated by dividing the subject's total number of consolations by its total number of opportunities to console

illustrates, we found a significant positive correlation between consolation tendency from earlier to later age-classes (Pearson's $r(20) = 0.61$, $P = 0.002$), indicating relative consistency over the lifespan. This relation corresponds to an intra-class correlation (ICC) of 0.61 (95% confidence interval: 0.27, 0.82; F(21, 21) = 4.1, $P < 0.001$).

Moreover, upon entering consolation tendency from younger age period(s) into the full GLMM, we also found it to be a significant predictor of consolation later in life ($z = 2.74$, $b = 46.19$, $P = 0.006$). This latter result reveals the robustness of this relation when controlling for qualities of the bystander-recipient relationship and other fixed effects known to influence the occurrence of consolation, which only this model could account for.

**Consolation and social competence**. Finally, we tested the relation between an individual's overall consolation tendency and its Composite Sociality Index (CSI). As shown in Fig. 3, we found a significant positive correlation (Pearson's $r(42) = 0.43$, $P = 0.003$), such that individuals who consoled more were more socially integrated. Furthermore, entering CSI as a predictor in the full GLMM also revealed a significant positive relation to consolation ($z = 4.39$, $b = 0.80$, $P < 0.001$).

**Discussion**
Here, we provide evidence for the relative consistency of individual variation in chimpanzee consolation behaviour, suggested to be a marker of empathy in humans and other animals[26]. The present study reveals that consolation tendencies exhibited moderate stability for up to 8 years, and perhaps most notably, across developmental stages, in a non-human species. Traditionally, variation in animal consolation behaviour has been explained by relationship quality rather than the potential role of stable individual differences. Although our findings corroborate prior research showing that valuable bystander-recipient dyads (as defined by affiliation and kinship) are most likely to console, our study illuminates that individual identity explains an additional, meaningful proportion of the variance in consolation behaviour. Importantly, individual variation in consolation could not be explained by variation in individuals' general tendency to

affiliate with others. That consolation was moderately stable over different recipients of aggression also provides evidence for a type of cross-situational consistency. Given the relative stability of individual differences in behaviour across time and context, together with previous similar findings regarding chimpanzee reconciliation behaviour[36], the results of the present study stress the need to include conflict management skills as a component of broader animal personality[37].

A key finding of our study is that individual variation in consolation behaviour is consistent across the full range of ontogenetic stages, from infants to adults. Being more prone to console distressed others in early life stages predicts higher consolation tendencies in older stages, which implies high persistence of individual differences over time. These findings parallel human research reporting that individual variation in prosocial and empathy-related responding has its origins in early childhood[18, 38, 39]. The stability of chimpanzee consolation behaviour may be explained by numerous factors, including genetic, physiological, developmental, ecological, maternal, or social factors, as is the case with human sympathetic concern[16, 40]. The recent findings that behavioural and dispositional empathy in humans[41, 42] and consolation behaviour in rodents[27] are oxytocin-dependent invites us to speculate that differences in the oxytocinergic system, in combination with other internal and external effects, might underlie both inter- and intra-species variation in empathetic responding. This possibility could be investigated in future comparative research by combining endocrinological, pharmacological, and observational methods.

It should be emphasized that the relative consistency of individual differences over time does not necessarily mean that an individual's tendency to console was constant across its lifespan. Actually, we found that, as in bonobos and gorillas[20, 23, 29], chimpanzees' tendency to provide consolation decreased with age. The decline in chimpanzee consolation behaviour across development is intriguing, as these findings challenge the assumption that consolation and related behaviours rest on advanced emotional/cognitive capacities that emerge and increase with age[14, 15] (see ref. [43] for an exception). These findings are especially provocative given that the majority of human studies in adults have used self-report methodologies[44], results of which do not always line up with performance-based empathy measures[45–47] (see, however, ref. [48]). Moreover, the psychometric

approaches applied to older human subjects differ considerably from the behavioural techniques applied to younger human subjects and animals, making generalizations difficult. A strength of the current study is that the developmental decline in consolation behaviour was found using the same behavioural measure over the lifespan. We should add, however, that this finding does not point to a drop in empathy-related responding per se, as it is possible that throughout development expressions of empathy become increasingly under cognitive control. In other words, advances in cognition could promote more filtered (for example, in-group biased[49]) manifestations of empathy.

This developmental decrease in consolation behaviour is also intriguing in light of the present finding that younger age-classes were no more likely to be the recipients of consolation than were older age-classes—thus, in contrast to adult individuals[24], immature individuals were not simply reciprocating the behaviour that other group members showed towards them. Conflicts can be costly, with renewed aggression making approaching a recent victim potentially risky. Perhaps, then, younger individuals are less prone to this risk (e.g., through the protection of older affiliates), a question that warrants future study. Another possibility is that younger group members are less discriminating in their social efforts. For example, a recent study revealed that Barbary macaques (*Macaca sylvanus*) become more selective with their social partners as they get older[50], allowing us to question whether chimpanzees' social networks, particularly when it comes to post-conflict behaviours, also become more refined over time (see ref. [51] for a review on socioemotional selectivity in humans).

An additional explanation can be gathered from the literature on Agreeableness and Extraversion. Whereas Agreeableness involves a sensitivity towards others, Extraversion involves the tendency to actively engage with conspecifics[30–35], highlighting that consolation could be a manifestation of both domains. As has been shown in humans and chimpanzees, Extraversion declines while Agreeableness increases with age[31, 34], with the chimpanzee study revealing that Extraversion declines were much larger than Agreeableness increases[35]. Thus, any measure that captures both domains might show a declining trajectory itself. Future studies should explore how consolation fits into these broader personality frameworks, and the extent to which age-related changes in this trait are linked to other individual difference measures. As has been done in the human research[32], it will also be interesting to examine whether the trait itself becomes increasingly stable across later age groups.

We also provide evidence that an individual's tendency to offer consolation to distressed others was highly correlated with its Composite Sociality Index, a sign of overall social competence. Our results, therefore, are in line with the notion that consolation and other abilities that may have an empathy basis facilitate other-oriented processes and behaviours, such as sharing, comforting, or helping, which in turn foster successful social relationships and better integration in social networks[2, 4]. For instance, people who report higher empathy are also more likely to help others, have stronger communication and conflict resolution skills, and richer social networks[17, 52, 53]. Similarly, juvenile bonobos' tendency to console mates is positively related to effective emotion regulation and social competence[23]. Our findings critically contribute to this literature by showing the latter association in our other closest primate relative, and by emphasizing aspects of social integration highly relevant to chimpanzee socio-emotional development (e.g., grooming and play). Further research exploring whether other aspects of social competence and related emotional skills predict consolation and other suggested markers of empathy (e.g., emotional contagion, helping and other prosocial behaviours) will greatly contribute to

a better understanding of the role of these capacities in animals' social lives.

**Table 2 GLMM for individual differences in consolation**

| Variable | Description | | |
|---|---|---|---|
| *Outcome*: | | | |
| Consolation | Occurrence of consolation w/recipient (0 = no; 1 = yes) | | |
| *Fixed effects*: | | | |
| Consolation opportunities | Bystander's # of consolation opportunities w/recipient | | |
| Bystander affiliation rate | Bystander's hourly affiliative rate w/all group members | | |
| Dyad affiliation level | Dyadic affiliation (0 = non-strong; 1 = strong) | | |
| Dyad kinship | Dyadic kin relationship (0 = non-kin; 1 = kin) | | |
| Bystander rank | Bystander's dominance rank (low; medium[a]; high) | | |
| Bystander sex | Bystander's sex (0 = male; 1 = female) | | |
| Bystander rank*sex | Bystander rank (0 = non-high; 1 = high) *sex (as above) | | |
| Bystander age | Bystander's age-class (infant[a]; juvenile; adolescent; adult) | | |
| Recipient rank | Recipient's dominance rank (low; medium[a]; high) | | |
| Recipient sex | Recipient's sex (0 = male; 1 = female) | | |
| Recipient age | Recipient's age-class (infant[a]; juvenile; adolescent; adult) | | |
| | | *Full model* | *Base model* |
| *Random effects*: | | | |
| Individual[b] | | Bystander X Recipient | Recipient |

[a]Reference groups
[b]Full model includes both bystander and recipient (using a crossed-random effects structure); base model includes only the recipient (using a regular random effects structure)

## Methods

**Subjects and housing**. Behavioural observations were conducted on 44 socially-housed chimpanzees at the Yerkes National Primate Research Center (YNPRC) in Atlanta, Georgia, USA. Two separate groups (FS1 and FS2) lived in large outdoor compounds (750 and 520 m², respectively) with access to heated indoor quarters. The compounds were equipped with a variety of climbing structures and enrichment items, with water and primate chow available ad libitum. The number of individuals per group varied slightly throughout the study period due to births, deaths or veterinary/management procedures, but at any time, both groups consisted of at least one adult male and several adult females. Subjects comprised all age-classes, including 15 infants at the onset of the observation period (Supplementary Table 1 for a detailed description of the study subjects). The YNPRC is accredited by the American Association for the Accreditation of Laboratory Animal Care, and all methods were approved by the Institutional Animal Care and Use Committee of Emory University.

**Data collection**. Since the establishment of the groups, 90-minute controlled observation sessions were conducted approximately once weekly by the same trained research technician, Mike Seres (described in further detail in ref. [54]). Between 1992–2000 for FS1 and 1994–2000 for FS2, all-occurrences of social interactions were recorded, including agonistic conflicts (defined by the presence of at least one of the following behaviours: tug, brusque rush, trample, bite, grunt-bark, shrill-bark, flight, crouch, shrink/flinch or bared-teeth scream[55, 56]). In the 10-minute period directly following aggression, all-occurrences of affiliation involving the former opponents were recorded, along with the timing, identities and initiators of those interactions. Additionally, scan samples of state behaviours (e.g., grooming, contact-sitting) were taken at regular intervals (every 5 min through 1993 and every 10 min in years thereafter).

We analyzed data from a total of 3003 agonistic conflicts (1676 in FS1; 1327 in FS2). Consistent with prior research, a bystander was an individual neither involved in the conflict nor in any aggressive interaction within 2 min before/after the conflict. Consolation behaviour was defined as the first affiliative contact directed from a bystander to the recipient of aggression during the post-conflict period (i.e., 10 min after the last exchange of agonistic behaviour between the opponents). While it is possible that some of the observed post-conflict affiliations

might be functionally different to consolation[25], results from previous analyses revealed that these affiliative contacts mainly function as consolation in our study groups[22, 24].

**Summary of statistical approach.** To assess the relative stability of consolation behaviour across the lifespan, we took a multipronged approach. First, we used generalized linear multilevel models (GLMMs) to estimate the effect of bystander identity on the probability of offering consolation to a recipient in a given time period controlling for the number of opportunities the bystander had to console the recipient. This approach allowed us assess the effect of the bystander over-and-above other control variables (e.g., the recipient's identity, the bystander's baseline affiliation tendency, and aspects of the bystander-recipient relationship). Second, we calculated subjects' consolation tendencies (number of consolations/number of opportunities) within each age-class on record and tested whether consolation tendency at an earlier age predicted consolation tendency at an older age, providing us with a metric of base repeatability of consolation behaviour across the lifespan. Finally, we generated each individual's overall consolation tendency (i.e., collapsed across all age-classes on record over the observation period) and compared this measure to general social integration/competence scores, which were also calculated for the entire observation period (detailed below). We also analyzed this relation using the GLMM collapsed by time period, which controlled for other factors such as rank and sex.

**Individual differences in consolation.** Individual differences in consolation behaviour were assessed by fitting a restricted maximum likelihood generalized multilevel model, using a binomial error distribution and logit link function[57, 58]. We tested the significance of individual variation by comparing two models, with and without bystander identity (Table 2 and below for details on model specification). Provided the fixed-effect structure remains the same, the additional explanatory power of adding one random effect to a model (in our case, bystander identity) can be measured using a log-likelihood ratio test[59]. Using this test to compare the fit of the full model to the base model, we determined whether bystanders' identities accounted for a significant portion of variance in consolation, and hence whether there were significant differences among them (the random intercept included in both models controls for repeated observations and tests for relative individual stability in response level). These models are known for their power and versatility, and as such, have become a common approach in animal personality research[60, 61].

Given prior research on consolation behaviour, our model structure needed to account for the quality of the relationship between bystander and recipient. To investigate the explanatory power of individual variation while simultaneously ruling out the bystander-recipient relationship as an alternative explanation, we made the dyad our unit of analysis. Data were structured per each possible bystander-recipient dyad per specified timeframe (FS1: 92–93, 94–96, 97-00 and FS2: 94–96, 97-00), reflecting periods where group composition remained relatively stable. The binary outcome of the model equalled whether (0/1) the bystander offered consolation to the recipient in the given time period. We then entered the number of opportunities the bystander had to offer consolation to the recipient within that period as a fixed effect (i.e., the number of the recipient's agonistic conflicts that the bystander witnessed in which the bystander was not involved). We also included a measure of the bystander's baseline tendency to affiliate by selecting, a posteriori, a total of 2645 (1482 in FS1; 1163 in FS2) control observations. Control observations were identical to post-conflict observations except that they were not preceded by any agonistic interaction during a period of at least 10 min. For each individual, a baseline affiliation score was calculated as the hourly rate of all affiliation given to any group member (including kiss, embrace, groom, gentle touch, finger/hand-in-mouth, mount, play, i.e., the same behaviours included as potential consolation behaviours).

Additional fixed effects included variables that have been previously shown to impact the occurrence of consolation in chimpanzees (Table 2). We included bystander-recipient affiliation, which was calculated via a combined measure of four state behaviours (contact-sitting, sitting within arm's reach, grooming and mutual grooming) collected during scan samples, using the quartile points of dyadic scores for each individual. Only dyads with scores in the top quartile were considered to have a strong affiliative relationship. We also included bystander-recipient kinship, which refers to matrilineal relationships, where only (grand)-mother-offspring and maternal siblings were considered kin (the one adoptive relationship was also considered kin). Dominance ranks of the bystander and recipient (calculated using the direction of submissive signals and non-agonistic approach/retreat interactions; see ref. [22] for details), as well as their (respective) sex and age-class, were also entered. Age-classes were defined as follows: infants (1–4 years old), juveniles (5–7 years old), adolescents (8–9 years old) and adults (10 years and above). Finally, we entered the interaction between the bystander's rank and sex, on the basis of previous findings in these groups revealing that high-ranking males were especially likely to offer consolation[24]. If we found an effect of a three-level factor (i.e., bystander and recipient rank) or four-level factor (i.e., bystander and recipient age-class) on the occurrence of consolation, we ran multiple comparisons between the groups to determine their relative effects in the full model. As we control for previously established patterns and explore new, specific predictions, corrections for multiple tests were not applied (see ref. [62] for further details).

**Repeatability of consolation over the lifespan.** A second set of analyses was conducted to further examine the relative repeatability of an individual's consolation tendency across the multi-year observation period. For this analysis, we used subjects who went through different age-classes during the observation period ($N = 22$). We compared consolation tendency from the youngest age period (s) on record to that of the oldest age period(s) on record for each individual. Age period refers to the entire timeframe in which a subject fell within a particular age-class. When data for more than two age-classes were available, the youngest age period corresponds to juvenile or to infant-juvenile grouped, and the oldest age period to adolescent or to adolescent-adult grouped. We then tested this relation using both a Pearson correlation and a GLMM. For the Pearson correlation, we calculated an individual's consolation tendency for the youngest and oldest age periods by dividing its total number of consolations by its total number of opportunities to console during each age period. We used this collapsed form of the data to calculate an ICC for consolation tendency across age periods (a single-measure, fixed raters model[63] implemented with the "ICC" package in R[64]). For the GLMM, we entered an individual's consolation tendency from the youngest age period(s) on record as a predictor of the probability of consolation occurring within the oldest age period(s) on record (controlling for opportunities) into the full model (Table 1).

**Consolation and social competence.** Finally, we generated a CSI to examine the relation between consolation and sociality, a sign of general social competence[65, 66]. Because grooming and maintaining proximity to other group members are widely considered to provide meaningful measures of social relationships among non-human primates including chimpanzees[67], we calculated the CSI from scan data using the hourly frequency of giving grooming, the amount of time giving grooming per observation hour, and the proportion of scan points in proximity with another individual. Beyond being the measures used by other researchers to generate CSIs, these were the four behaviours of our subjects that showed the highest inter-correlation values across all periods and for each period independently. However, because this is not a suitable index of sociality for immature individuals (who spend far more time playing than grooming), an independent CSI was calculated for infants/juveniles using play behaviour (Supplementary Fig. 1 for developmental curves justifying this approach). All mother-infant interactions were excluded from the database to calculate the CSI index.

To calculate the CSI $\left(\text{CSI} = \sum \frac{x_i}{m_i}/4\right)$, the sum of each mean affiliative value per individual ($x_i$) was divided by the group median ($m_i$), which was then divided by the total number of behavioural measures in the analysis (i.e., 4 for adolescents/adults, 1 for infants/juveniles). This measure thus indicates the degree to which each individual deviates from the group on all four measures combined (their degree of sociality). High CSI values represent individuals who are more socially integrated than the median individual, low CSI values represent individuals who are less socially integrated than the median individual.

**Data availability.** The authors declare that the data supporting the findings of this study are available from the corresponding authors on request.

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

## Acknowledgements

We would like to thank Michael Seres and Filippo Aureli for the behavioural data collection of this study, and the animal care and veterinary staff at the Yerkes National Primate Research Center (YNPRC) for maintaining the health and wellbeing of the chimpanzees. This work was supported by a National Institutes of Health National base grant to the YNPRC (RR-00165; currently supported by the Office of Research Infrastructure Programs/ODP51OD11132), Emory University's College for Arts and Sciences, and the Living Links Center.

## Author contributions

C.E.W. and T.R. designed the study and wrote the paper; C.E.W. and B.F. conducted the data analyses; F.B.M.d.W. provided long-term data and grant support. All authors contributed feedback to and edited the manuscript.

## Additional information

**Competing interests:** The authors declare no competing financial interests.

