## [Peer Review File · Nature Communications]

Reviewer #1 (Remarks to the Author):

Review of NCOMMS-16-22607: Empathic Personalities: Chimpanzees Consistently Vary in the Tendency to Console Others

In their manuscript the authors present a series of analyses of behavioral observations from two groups of chimpanzees (N=44) who had been observed over a ~9- and 7~year period. The behavioral data included consolation behavior, a purported measure of empathy, and a "composite sociality index," which was used to measure the degree to which a chimpanzee was socially competent. They found that consolation behavior was stable over time (repeatable), associated with social competence, and that, contrary to what would be expected based on human findings, that older chimpanzees had lower scores than younger chimpanzees. They interpreted these findings as being informative with respect to the evolution of human empathy.

Overall, I liked what the authors were trying to do here. Specifically, I wholeheartedly agree with their statement that "The results of the present study stress the need to include conflict management skills as a component of broader animal personality." (p. 10) In fact, this should be the key focus of the paper. In addition, it's worth noting that the study had several strengths, including the fact that it was a longitudinal study and based on a lot of detailed observational data. On the other hand, the manuscript had some shortcomings, too, which tempered my enthusiasm for the manuscript, and which I detail below.

1. The authors need to better set out and tighten up the study rationale. Presently, it seems scattered. Moreover, as this is ostensibly a study about personality in chimpanzees, but also humans, they need to bring the human personality literature into their Introduction. Specifically, the human personality trait most closely related to the trait they seek to study would be the Agreeableness domain that is found in the five-factor model (McCrae and Costa 1991, Costa and McCrae 1995) and in the HEXACO model (Ashton and Lee 2007). Similar dimensions are found in other personality models, too, but given that these two are the major models being looked at today, there is probably no need to talk about these traits in other personality models. In addition, the authors should cite the longitudinal work on Agreeableness and its facets (Terracciano et al. 2005) as well as on related traits (for reviews, please see Roberts and DelVecchio 2000, Roberts et al. 2006). Along with this the authors should mention that personality dimensions like Agreeableness, which captures traits related to empathy and consolation, have been identified in large rating-based studies of chimpanzees across multiple studies and settings (King and Figueredo 1997, King et al. 2005, Weiss et al. 2007, Weiss et al. 2012, Freeman et al. 2013). There is also evidence for the stability of Agreeableness and for its tendency to be higher in older chimpanzees (King et al. 2008). Please note that I do not want the authors to clutter their manuscript by citing all of these studies, but they should at least cite the human literature on Agreeableness and the King et al. (2008) study, the latter being directly relevant to their hypotheses.

2. I had some problems with some analyses and how they were presented. First, the decision to focus on the correlation between consolation at younger and older age groups seems ad hoc. The authors should follow the accepted standard in the animal personality literature (Bell et al. 2009, Nakagawa and Schielzeth 2010) and compute the intraclass correlation using all of their observations. Naturally, as it is of interest in the animal (Bell et al. 2009) and human (Roberts and DelVecchio 2000) literatures, the authors could, in addition to examining their whole sample, split their sample into different age categories and examine whether the trait becomes increasingly stable across later age groups.

I also had a question about some analyses described on page 15. The authors provided their formula for computing the "relative repeatability" but do not cite the relevant formula number or a page number in Nakagawa and Schielzeth (2010) or whether they used some R package to conduct the analyses. I was thus left wondering about the formula because it appears to be

incorrect. What the authors appear to be examining here (it's a really interesting question) is whether consolation is consistent over different recipients of aggression (a sort of cross-situational stability). If so, then the denominator should be the sum of the variance due to bystander ID + variance of the error term, which would be the bystander*recipient interaction (relevant papers, including with applications to animals, are Shavelson et al. 1989, Shavelson and Webb 1991, Figueredo et al. 1995, Hernández-Lloreda and Colmenares 2006). I would like to add to this that, if possible, it would be extremely interesting to examine whether there is consistency in consolation behavior across different aggressors.

Another problem I identified is that, when reporting p-values, the authors should provide more detail throughout. A good guide that I like to follow comes from some medical journals, e.g., *Psychosomatic Medicine*: "If $p < .10$, then it should be expressed to 3 digits after the decimal point. If the value is $.10$ through $.99$, then it should be expressed to 2 digits. Values of $.000$ and 1.0 should be reported as $< .001$ and $> .99$, respectively." Moreover, the authors report a $p=.103$ as indicating that infants are "marginally more likely to provide consolation than juveniles". It is now recognized that, as p-values are uniformly distributed under the null hypothesis, talk of effects being "trends", "borderline significant", etc. is inappropriate (for additional problems with this practice see Wood et al. 2014).

A final note with respect to the methods and results: there is some detail missing. For example, concerning multiple comparisons (p. 6) the authors need to note whether these comparisons corrected for multiple tests and, if so, what type of correction was applied. Also, were any variables standardized? If so, the authors should note this and be cautious about whether a b or beta is reported as the latter is a standardized coefficient and the former is not.

3. With respect to their Discussion, the authors claim that the development of empathy in chimpanzees is the opposite to what it is in humans. I do not believe that their data shows this nor do I believe that this difference could be attributable to the use of self-report measures in many human studies. Starting with the latter claim, studies that use peer-ratings find evidence for the same developmental differences in many cultures (McCrae et al. 2005), studies of chimpanzee ratings find that Agreeableness increases, and finding in humans that use behavioral observations also report that older people are higher in empathy (Richter and Kunzmann 2011). What I suspect is going on is that the apparent difference in human and chimpanzee age trajectories can be explained by the well-known problem of identifying what animal personality trait or traits are measured by one or more behavioral measures (Carter et al. 2012a). For example, two seemingly similar measures of behavior in chacma baboons measure two different traits (Carter et al. 2012b). As such, I suspect that consolation behavior is a manifestation of high Agreeableness and high Extraversion, i.e., it is a measure of tendency to feel empathy and the tendency to interact with conspecifics or act on those feelings. Because in chimpanzees, Extraversion declines are much larger than Agreeableness increases, any measure that assesses both domains would show a decline. Evidence for this explanation is in the present results: although human women are higher in Extraversion than men, the reverse is true in chimpanzees (e.g., King et al. 2008). With this in mind I checked the direction of sex differences for consolation behavior. As I predicted, female chimpanzees were lower than males (see Table 1).

To address this, the authors should raise the above explanation as a possibility and drop the discussion about self-reports as a possible explanation, which, frankly, adds little to the paper as many of these concerns about self-reports have long been shown to be groundless. I would also like to encourage them to come up with a way to test this possibility and to think about what it would mean if the developmental trends are similar.

In addition to these more substantive comments, I had several minor suggestions.

Page 2

Please note the sample size, length of follow-up times, and number of observations in the Abstract.

I think the phrase "empathetic personalities" is awkward. The phrase "individual differences in empathy" is a bit better.

I think "empathy driven response" should simply be "traits".

Please note what the "insights" from this study were?

Page 3

Please change "Not only does it" to "Not only does empathy". I think the "but" in this sentence can be deleted.

I would delete the word "accumulating". Also, variations of this sentence are found throughout the animal personality literature. It would be nice to see a new way to start.

I would delete the word "facets" as it is not really the same as "levels" in this context.

What are the "empathetic tendencies" that are found early in life?

I think the word "both" in the center of the page can be deleted.

The phrase "Despite these changes" is an awkward way to start the sentence. I think the word "also" can be deleted in the same sentence.

There is no need to italicize "individual differences".

The phrase "conceptualized as part of a broader prosocial personality" is awkward. Do you mean personality dimension or domain?

Page 4

I would change "remain hard to elucidate" to "are yet to be elucidated".

The reference here to the oxytocin literature should be a bit uncritical, particularly given that the human findings related to oxytocin are wanting (Nave et al. 2015, Lane et al. 2016).

I do not think the sentence "However, these studies failed to..." is necessary.

Again, the italics here are not needed.

Page 5

I would change "also exist" to "exist".

Here and elsewhere, I would change "a number of" to "numerous", "several", "many", or the like.

I think the parenthetical material that follows "Finally" is probably better separated off with commas.

Here again, what "novel insights" are you expecting?

I would delete "a variety of".

Here you insert "see" before calling out a table whereas you do not do so elsewhere.

Page 6

I would delete the word "importantly".

.052 is quite small. I wonder if this is because of the possible problem with how this was computed (see my earlier comments).

Page 7

There should be error bars in the figure. Also, in the table (and in other tables), I would substitute "Test Statistic" with the statistic (z in the present case). The p-values in the table that are below .001 should be "< .001" and not ".000".

Page 8

Does "Figure 1" here (first paragraph) actually refer to Figure 2? I could not find any call out for Figure 2 and it seems to be more consistent with the results described here.

Page 9

Please change "relation" to "relationship".

I do not think that Figure 3 is cited in the text.

Page 10

Please avoid so-called statements of priority. Just because a study is first does not mean it is interesting or important.

I would delete "distinct" here.

Page 11

I would delete "and/or temperament" as the distinction between temperament and personality is arbitrary. Please note, too, that environmental factors can also be individual as is the case with the nonshared environment that behavior geneticists find to be important.

I think the speculation here goes too far, particularly given the question of whether the differences are, in fact, present (see my earlier comments). However, the point about needing to compare these systems in humans and chimpanzees is an excellent one.

Page 12

It is not clear to me what the authors are trying to say in their sentence that begins "We should add..." I suggest revising this to make it clear.

In the first full paragraph, what result does "This result" refer to?

The point made in the sentence "Another possibility is that younger..." is excellent. I recall hearing from a colleague some time ago that there is a human literature on this kind of thing, too, though I cannot remember any specifics. I would thus encourage the authors to see if they can identify at least one relevant study or review in humans.

Page 13

The authors here begin talking about "emotional competence" which was not raised earlier. I think they should stick to the terminology they used throughout.

The four parenthetical statements that begin "e.g." are a distraction. I would revise so as to avoid this.

I think "more detailed" can simply be "detailed".

Page 14

I would change "behavioural elements" to "behaviours".

Page 15

I think the authors can delete "Specifically" in this instance.

The sentence that begins "Provided the fixed..." is a bit awkward and too long.

The sentence that begins "Using this test..." is also too long and it repeats information from earlier.

Supplementary Figure 1

Here it would be worth spelling out what FS1 and FS2 mean. I know it is in the text, but it should be in the figure legend.

Thank you for asking me to review this interesting manuscript. I hope the authors find my comments helpful.

Alexander Weiss [I sign my reviews]

References

- Ashton MC, Lee K (2007) Empirical, theoretical, and practical advantages of the HEXACO model of personality structure. *Personality and Social Psychology Review* 11:150-166
- Bell AM, Hankison SJ, Laskowski KL (2009) The repeatability of behaviour: a meta-analysis. *Anim Behav* 77:771-783
- Carter AJ, Feeney WE, Marshall HH, et al. (2012a) Animal personality: What are behavioural ecologists measuring? *Biological Reviews* 88:465-475
- Carter AJ, Marshall HH, Heinsohn R, et al. (2012b) How not to measure boldness: Demonstration of a jingle fallacy in a wild social primate. *Anim Behav* 84:603-609
- Costa PT, Jr., McCrae RR (1995) Domains and facets: Hierarchical personality assessment using the Revised NEO Personality Inventory. *J Pers Assess* 64:21-50
- Figueredo AJ, Cox RL, Rhine RJ (1995) A generalizability analysis of subjective personality assessments in the stump-tail macaque and the zebra finch. *Multivar Behav Res* 30:167-197
- Freeman HD, Brosnan SF, Hopper LM, et al. (2013) Developing a comprehensive and comparative questionnaire for measuring personality in chimpanzees using a simultaneous top-down/bottom-up design. *Am J Primatol* 75:1042-1053
- Hernández-Lloreda MV, Colmenares F (2006) The utility of generalizability theory in the study of animal behaviour. *Anim Behav* 71:983-988
- King JE, Figueredo AJ (1997) The Five-Factor Model plus Dominance in chimpanzee personality. *J Res Pers* 31:257-271
- King JE, Weiss A, Farmer KH (2005) A chimpanzee (*Pan troglodytes*) analogue of cross-national generalization of personality structure: Zoological parks and an African sanctuary. *J Pers* 73:389-410
- King JE, Weiss A, Sisco MM (2008) Aping humans: Age and sex effects in chimpanzee (*Pan troglodytes*) and human (*Homo sapiens*) personality. *J Comp Psychol* 122:418-427
- Lane A, Luminet O, Nave G, et al. (2016) Is there a publication bias in behavioural intranasal oxytocin research on humans? Opening the file drawer of one laboratory. *J Neuroendocrinol* 28
- McCrae RR, Costa PT, Jr. (1991) Adding Liebe und Arbeit: The full Five-Factor Model and well-being. *Pers Soc Psychol Bull* 17:227-232
- McCrae RR, Terracciano A, 78 Members of the Personality Profiles of Cultures Project (2005) Universal features of personality traits from the observer's perspective: Data from 50 cultures. *J Pers Soc Psychol* 88:547-561
- Nakagawa S, Schielzeth H (2010) Repeatability for Gaussian and non-Gaussian data: A practical guide for biologists. *Biological Reviews* 85:935-956
- Nave G, Camerer C, McCullough M (2015) Does oxytocin increase trust in humans? A critical review of research. *Persp Psychol Sci* 10:772-789

Richter D, Kunzmann U (2011) Age differences in three facets of empathy: Performance-based evidence. *Psychol Aging* 26:60-70

Roberts BW, DelVecchio WF (2000) The rank-order consistency of personality traits from childhood to old age: A quantitative review of longitudinal studies. *Psychol Bull* 126:3-25

Roberts BW, Walton KE, Viechtbauer W (2006) Patterns of mean-level change in personality traits across the life course: A meta-analysis of longitudinal studies. *Psychol Bull* 132:1-25

Shavelson RJ, Webb NM (1991) Generalizability theory: A primer. Sage, Thousand Oaks, CA

Shavelson RJ, Webb NM, Rowley GL (1989) Generalizability theory. *Am Psychol* 44:922-932

Terracciano A, McCrae RR, Brant LJ, et al. (2005) Hierarchical linear modeling analyses of the NEO-PI-R scales in the Baltimore Longitudinal Study of Aging. *Psychol Aging* 20:493-506

Weiss A, Inoue-Murayama M, King JE, et al. (2012) All too human? Chimpanzee and orang-utan personalities are not anthropomorphic projections. *Anim Behav* 83:1355-1365

Weiss A, King JE, Hopkins WD (2007) A cross-setting study of chimpanzee (*Pan troglodytes*) personality structure and development: Zoological parks and Yerkes National Primate Research Center. *Am J Primatol* 69:1264-1277

Wood J, Freemantle N, King M, et al. (2014) Trap of trends to statistical significance: likelihood of near significant P value becoming more significant with extra data. *Br Med J* 348:g2215

Reviewer #2 (Remarks to the Author):

The authors report an interesting study on consistent variation regarding consolation in chimpanzees. They analyzed a very extensive long-term (up to 8 year) behavioural dataset of a total of 44 chimpanzees and found that the tendency to console others after a conflict varies significantly among those individuals, and that this variation is consistent over a long period of time / age-classes. Moreover, they show that even though this variation remains consistent, the general tendency to console decreases with age in chimpanzees. I think this is an interesting, well performed study. However, I have one major conceptual concern with the study that puts the novelty of this study into question, and therefore, I feel this study doesn't warrant publication in *Nature Communications* and would better fit a more topical journal.

Major concern:

The authors claim that they have found empathic personalities, which is based on the premise that consolation has empathic motivations. However, this does not necessary need to be the case in chimpanzees, as has been nicely reviewed by a paper by Fraser and colleagues (2009 *Communicative & Integrative Biology*), which the authors do not cite. Some studies have shown that what is generally referred to as post-conflict 3-rd party affiliation, can also be a way to avoid redirected aggression. As such, rank effects can also play an important effect in for example the opportunity to console, which is an important factor in calculating the variation in consolation in this manuscript.

Given that I'm not convinced that the behaviour the authors report here equals empathy, we're left with a study that shows long-term consistency of a behaviour; i.e. personality, which is not very novel.

Although I applaud the authors for taking the observational approach, it should also be noted that personality studies on chimpanzees that use the questionnaire approach use 'Conciliatory' as one of the adjectives describing their subjects (see for the original adjective list Goldberg 1990 JSPS) and also show consistency in its overarching construct (personality factor) Agreeableness in chimpanzees, and some studies also report long-term consistency (King et al. 2008; JCP). Also that latter study reports aging effects on personality traits and actually reports an opposite effect; i.e. Agreeableness increase with age, albeit only slightly. Again this paper is not, yet should be cited.

Minor comments:

throughout: don't say consolation is empathy driven and actually refer to it as post-conflict 3-rd party affiliation. I'm fine with speculation, yet don't take it as a truth.

I. 47. ALL animals??

I. 58. please provide refs

I. 61. ref 20, 23. You should be cautious with referring to the age effects in these studies since the sample of Bonobos in these studies contained very few older individuals (i.e. only 8 older than or equal to 15, and only 2 older than or equal to 20).

I. 105-112. Did you perform any correction for multiple comparisons?

I. 120. Please mention in the table what where the reference categories.

I. 122. To calculate repeatability, ICC's are more appropriate as they specifically take inter-individual variation into account (see Lessels & Boag 1987 AUK)

I. 253. Please mention how age-classes are defined

I. 275. What were the reference groups for Affiliation, Kinship, Sex, and recipient sex?

I. 294. why did the authors choose a binary way of analysing affiliative relationships rather than putting it in as a continuous variable, if you have the data anyway?

I. 312. What is an age-period. I think this part needs a bit more explanation.

Reviewer #3 (Remarks to the Author):

The paper reports individual stability in the tendency to offer post-conflict consolation in captive chimpanzees. The study takes advantage of the large data set on post-conflict behaviour in the records of the Yerkes chimpanzees. The aim taps nicely into the hot topic of behavioural consistency vs. plasticity (i.e. personality) and assesses it in the realm of social behaviour. The results are interesting, given that we still know little of the mechanisms of consolation and because personality in social behaviour is still relatively poorly studied. The study is likely to prompt further research into the personality effects on social behaviour, and into the factors influencing socio-positive and prosocial behaviour.

However, I have some issues with the paper with regards to the methods and the scope.

One of my concerns is about the way the study assesses consolation. As we know by now, not all bystander-initiated affiliative contacts to a conflict victim necessarily serve the consolatory function. It is commonplace to lump them, but it turns out that there are several functions to the seemingly similar-looking behaviour (Fraser et al. 2009). I do realise that in this kind of a data set it is not possible to pull these functionally different behaviours apart. However, I would appreciate at least an acknowledgment of this caveat in the introduction.

Secondly, the data were never collected to study post-conflict behaviour. Therefore, a major caveat is that there is no way of controlling the effect of individuals' baseline affiliation to partial it out from 'true' consolation. In controlled studies, this is done with well-established methods that yield a 'corrected consolation tendency' for each individual. This is not possible to derive for this study. Therefore, there is inevitably the possibility that individuals' post-conflict affiliation tendency is influenced by their general affiliation tendency. However, the group has published a number of

papers on consolation with this dataset and acknowledged this caveat in their earlier papers. A note on this would be needed here, too, at least as a discussion point.

Both of these issues are, in my opinion, highly relevant in the discourse of consolation behaviour, but given the weight of the group's earlier papers with this same dataset, I suppose we just have to agree to disagree, and move on.

That said, the following issues should be taken into account in the revision.

- The paper is framed in line with personality literature and the title suggests empathic tendency as a personality trait. Yet, the background of and implications on personality literature are very thin. As links are drawn to the stability of human empathic tendency, the literature on human personality is highly relevant, but totally absent in the paper. Similarly, there is no introduction into animal personality in general or chimpanzee personality in particular. Since there is a large body of research on primate personality available by now, this needs to be tied into the paper.

- The crucial metric to establish 'personality', or relative within-individual consistency in behaviour, is repeatability. In this study repeatability is calculated differently from the usually recommended way in the personality literature. It is relevant to establish not only the amount of variance in one individual's repeated observations but also, in relation to that, the variance among individuals. As far as I understand, the formula on l. 252 does not give that. Please calculate in addition repeatability following Lessells & Boag 1987. It gives a value between 0 and 1, with higher repeatability indicating small contribution of within-individual variance to the overall variance and, thus, allows one to estimate in a direct and intuitively understandable way the degree of within-individual consistency and among-individual differences.

Specific comments

Introduction

- l. 31: The references here include papers on human empathy, mice prosocial behaviour, and three review papers. As far as I know, these papers don't report the facilitating effect of similarity on empathy in animals. The social closeness effect is shown in primates and the familiarity effect is shown in mice.

- l. 41-43. Given that your paper is framed in line with personality literature and the title suggests empathic tendency as a personality trait, the literature on human personality and animal personality are both highly relevant. The studies cited here show nicely the developmental aspects of empathy and prosociality, but why is literature on the association of empathy to human personality structure and the developmental effects thereof ignored? Empathy and prosociality are associated with Agreeableness, one of the five constructs in human personality according to the most widely used psychometric model (the Five Factor Model). Agreeableness shows age-related increase in humans (e.g. McAdams & Olson 2010), and there is a study reporting the same in chimpanzees (albeit with a human-psychology derived method of assessing personality; King et al 2008). This discussion is very relevant to your paper, since you want to establish empathic tendency as a personality trait.

- l. 45-46. Related to the above comment, that is not really true when you consider the human personality research in general. It's also relevant to clarify here what is meant by consistency: a behavioural tendency that does not change or tendency that allows change but shows (relatively high) rank-order stability.

-l. 53ff. As you note, in addition to oxytocin, HPA-axis is known to be involved. This was nicely demonstrated in a recent study on rats: anxiolytic drug blocked helping behaviour that was likely be empathy-driven. Worth citing here? (Ben-Ami Bartal et al 2016)

Methods:

- l. 255 "metric base of repeatability". This does not really give a metric that would be comparable to repeatability in animal personality studies.

- l. 328-320 and l. 234 ff: I don't find this in Table 1. Based on the model description earlier, I thought that Table 1 shows the occurrence of consolation as the response variable and has dyad as a unit of analysis. Where is the model for individual's given consolation in an older age class as the response variable? The results section on this is equally unclear. Given the centrality of this result to your argument, this needs to be clarified.

- The approach specified on l. 260 ff is appropriate to establish the amount of variance that is explained by among-individual variation, so no issues there. However, I still argue for the value of establishing the classic sense repeatability values for individuals' consolatory tendency, because as a generally used metric it allows an easy and intuitive way of assessing the relative consistency of behaviour.

Results

- l. 97: this perfectly illustrates the difficulty with using only a mixed model with other predictors as the only metric for repeatability. $r=0.052$ is a really low figure, but it obviously is directly influenced by the model specifics. So pls see my above comment on calculating repeatability (e.g. with intra-class correlation coefficient, see e.g. McGraw & Wong, 1996).

- l. 105ff: The betas of adolescents and adults here are missing the negative sign (provided they are correct in the Table 1). The text is also somewhat confusing in wording, it's not always clear which group the stats in parentheses refer to.

- l. 122 ff: Not Fig 1 but Fig 2, I suppose.

- l. 140 ff: Why did you not run a model of this one? I would think that individual's CSI is also influenced by rank and sex, so having CSI as a response variable and sex, rank, age, and consolation tendency as predictors would nicely show its relative predicting value.

Discussion

- l. 158-159: here a broader incorporation into the literature of animal personality is pertinent. How does consistency in this form of social behaviour potentially relate to other findings on chimpanzee (social) personality traits? What would you predict for other social species?

- l. 167. The mechanisms maintaining relative consistency in behaviour are indeed numerous, and there is great literature on this in the animal personality field. The suggestions outlined here though, "genetic, physiological, and/or temperamental", are all aspects of the same, i.e. intrinsic factors that regulate the outcome behaviour is individual's dispositional behavioural tendency. Here, a more appropriate discussion would outline the relative effects of internal and external effects and the potential interactions thereof. Things like maternal effects, developmental effects, the effects that shape the outcome behaviour in a short term (such as facilitation), and the effects of the learning environment are all known to influence personality. Of course, oxytocinergic system is probably at play here, but the outcome behaviour is a result of internal and external effects combined. This could be elaborated with using the recent studies on such effects on personality in other species.

- l. 147: The wording is strange here. Yes it is 'constant' but behavioural consistency refers to relative stability and rank-order stability. The fact that consolatory tendency declines with age but individuals are relatively stable (based on Fig 2) suggests that there is indeed rather high rank-order stability in your data.

- l. 190: Were infants the targets of aggression frequently enough to allow this inference?

References

- Fraser, O. N., Koski, S. E., & Wittig, R. M. (2009). Why are bystanders friendly to recipients of aggression? *Communicative & Integrative Biology* 2(3), 285-291.
- Lessells CM, Boag PT. 1987. Unrepeatable repeatabilities: a common mistake. *AUK* 104:116-121.
- McGraw KO, Wong SP. 1996. Forming inferences about some intraclass correlation coefficients. *Psychol Methods* 1:30-46.
- McAdams, D. P., & Olson, B. D. (2010). Personality Development: Continuity and Change Over the Life Course. *Annual Review of Psychology*, 61(1), 517-542.
- King, J. E., Weiss, A., & Sisco, M. M. (2008). Aping humans: Age and sex effects in chimpanzee (*Pan troglodytes*) and human (*Homo sapiens*) personality. *Journal of Comparative Psychology*, 122(4), 418-427.
- Ben-Ami Bartal, I., Shan, H., Molasky, N. M. R., Murray, T. M., Williams, J. Z., Decety, J., & Mason, P. (2016). Anxiolytic Treatment Impairs Helping Behavior in Rats. *Frontiers in Psychology*, 7(185), 19-14.

Dear Dr Webb,

Your manuscript entitled "Empathic Personalities: Chimpanzees Consistently Vary in the Tendency to Console Others" has now been seen by three referees, whose comments are appended below. You will see from their comments copied below that while they find your work of potential interest, they have raised quite substantial concerns that must be addressed. In light of these comments, we cannot accept the manuscript for publication, but would be interested in considering a revised version that addresses these serious concerns.

We hope you will find the referees' comments useful as you decide how to proceed. Should further analyses and text revision allow you to address these criticisms, we would be happy to look at a substantially revised manuscript. However, please bear in mind that we will be reluctant to approach the referees again in the absence of major revisions. If the revision process takes significantly longer than three months, we will be happy to reconsider your paper at a later date, as long as nothing similar has been accepted for publication at Nature Communications or published elsewhere in the meantime.

We are committed to providing a fair and constructive peer-review process. Do not hesitate to contact us if you wish to discuss the revision or if there are specific requests from the reviewers that you believe are technically impossible or unlikely to yield a meaningful outcome.

When resubmitting your paper, we also ask that you ensure that your manuscript complies with our editorial policies. Specifically, please ensure that the following requirements are met, and any relevant checklists are completed and uploaded with the revised article:

Reporting requirements for life sciences research: http://www.nature.com/article-assets/npg/ncomms/authors/ncomms_lifesciences_checklist.pdf

Please use the following link to submit your revised manuscript, point-by-point response to the referees' comments (which should be in a separate document to any cover letter) and any completed checklist:

<http://mts-ncomms.nature.com/cgi-bin/main.plex?el=A6S5oNI7A1GZfC5I2A9ftd45gNJpP5YLHQ8OZbjT69wZ>

Please do not hesitate to contact me if you have any questions or would like to discuss the required revisions further. Thank you for the opportunity to review your work.

Best regards,

Alexa McKay, PhD
Associate Editor
Nature Communications

Thank you again for the opportunity to revise our work. Please see below for our detailed responses to the reviews. All three reviewers' comments coincided in several areas, which allowed us to tackle their concerns simultaneously in the manuscript itself. However, below we address each of their comments separately.

Reviewers' comments:

Reviewer #1 (Remarks to the Author):

Review of NCOMMS-16-22607: Empathic Personalities: Chimpanzees Consistently Vary in the Tendency to Console Others

In their manuscript the authors present a series of analyses of behavioral observations from two groups of chimpanzees (N=44) who had been observed over a ~9- and 7~year period. The behavioral data included consolation behavior, a purported measure of empathy, and a "composite sociality index," which was used to measure the degree to which a chimpanzee was socially competent. They found that consolation behavior was stable over time (repeatable), associated with social competence, and that, contrary to what would be expected based on human findings, that older chimpanzees had lower scores than younger chimpanzees. They interpreted these findings as being informative with respect to the evolution of human empathy.

Overall, I liked what the authors were trying to do here. Specifically, I wholeheartedly agree with their statement that "The results of the present study stress the need to include conflict management skills as a component of broader animal personality." (p. 10) In fact, this should be the key focus of the paper. In addition, it's worth noting that the study had several strengths, including the fact that it was a longitudinal study and based on a lot of detailed observational data. On the other hand, the manuscript had some shortcomings, too, which tempered my enthusiasm for the manuscript, and which I detail below.

Thank you for your interest in our research and moreover your appreciation for its novelty and importance. While we purposely kept our focus broad enough to appeal to the journal's interdisciplinary readership, taking your suggested edits inherently shifted our focus more towards post-conflict skills as a component of animal personality. We would also like to note that this topic has already been a key focus of two other recent papers, which are both cited in the manuscript (in the above statement [lines 178-181]).

1. The authors need to better set out and tighten up the study rationale. Presently, it seems scattered. Moreover, as this is ostensibly a study about personality in chimpanzees, but also humans, they need to bring the human personality literature into their Introduction. Specifically, the human personality trait most closely related to the trait they seek to study would be the Agreeableness domain that is found in the five-factor model (McCrae and Costa 1991, Costa and McCrae 1995) and in the HEXACO model (Ashton and Lee 2007). Similar dimensions are found in other personality models, too, but given that these two are the major models being looked at today, there is probably no need to talk about these traits in other personality models. In addition, the authors should cite the longitudinal work on Agreeableness and its facets (Terracciano et al. 2005) as well as on related traits (for reviews, please see Roberts and DelVecchio 2000, Roberts et al. 2006). Along with this the authors should mention that personality dimensions like Agreeableness, which captures traits related to empathy and consolation, have been identified in large rating-based studies of chimpanzees across multiple studies and settings (King and Figueredo 1997, King et al. 2005, Weiss et al. 2007, Weiss et al. 2012, Freeman et al. 2013). There is also evidence for the stability of Agreeableness and for its tendency to be higher in older chimpanzees (King et al. 2008). Please note that I do not want the authors to clutter their manuscript by citing all of these studies, but they should at least cite the human literature on Agreeableness and the King et al. (2008) study, the latter being directly relevant to their hypotheses.

We very much agree here; the human and nonhuman primate research on Agreeableness were both essential additions to our manuscript. With respect to this, in the revision, we have incorporated this personality dimension in the manuscript's introduction [79-87] and discussion [224-234]. Our changes to the discussion are outlined in more detail below, in reply to your point #3.

In the introduction, we now reference the human literature on Agreeableness as a personality domain that features in widely used psychometric models. At your suggestion, we cite longitudinal work indicating its relative consistency and age-related increases in humans. Further, we now incorporate the work on this domain in chimpanzees, including the King et al. (2008) study, which we appreciate is highly relevant. We also took the opportunity to highlight the uniqueness of our study in assessing consolation (i.e., a behavioral measure) rather than using a rating-based methodology, although we believe both approaches are useful.

Covering the empathy, human personality, and animal consolation literatures simultaneously was a challenge—but now we believe our attention is more balanced. We hope that this has served to make our overall study rationale more cohesive and convincing. Thank you for your suggestions and for providing us with many key supporting references. They have certainly enhanced our presentation and perspective.

2. I had some problems with some analyses and how they were presented. First, the decision to focus on the correlation between consolation at younger and older age groups seems ad hoc. The authors should follow the accepted standard in the animal personality literature (Bell et al. 2009, Nakagawa and Schielzeth 2010) and compute the

intra-class correlation using all of their observations. Naturally, as it is of interest in the animal (Bell et al. 2009) and human (Roberts and DelVecchio 2000) literatures, the authors could, in addition to examining their whole sample, split their sample into different age categories and examine whether the trait becomes increasingly stable across later age groups.

Thank you for your clear suggestions regarding our statistical models and ways to ensure that our data analyses are consistent with the human and animal personality literatures.

Correlation between consolation at younger and older age groups: Because of the complex nature of our research questions and structure of the data, no single model would suffice to provide a complete picture of the consistency (or inconsistency) of consolation behavior. Thus, before embarking on data analysis, we decided to use a multiple-model approach. Reconfiguring the data and applying different analyses can help readers gauge the robustness of the effect. Furthermore, given the potential importance of our results, we wished to examine the effect from as many angles as possible (without redundancy) because it was entirely possible that one particular analysis approach could have found a lack of consistency. Had this been the case, it would have been necessary to consider why the chimps appeared to be consistent in their consolation tendency from one modeling perspective, but inconsistent from another perspective. Thus, part of the reason for our inclusion of the correlation between younger and older age groups was to test the trustworthiness and extent of the effect. As we find similar effects across all of our very different analysis approaches, we provide evidence for its robustness.

The specific reason for including the correlation is its simplicity. Generalized multilevel models are complex, could be unfamiliar to some readers, are still subject to research themselves (e.g., Allegate et al., 2016), and contain multiple messages in a single model. In contrast, correlations are familiar and direct; ours communicates the straightforward message that consolation tendencies in chimpanzees are remarkably stable over long periods of time. As such, we find them to make a valuable and central contribution to our results.

Intra-class correlations: We completely agree on the importance of following field standards for data analysis practices, which is why we used the Nakagawa and Schielzeth (2010) paper you mention in your comment as our primary data analysis guide. They write, “R [repeatability] is often called the intra-class correlation coefficient (ICC)” (Nakagawa and Schielzeth, 2010; page 937). Thus, when we reported repeatabilities in our manuscript, we were in fact reporting ICCs. However, we wanted to go a step farther and assess whether we found consistency in consolation behavior even after controlling for (i) confounding factors such as sex, age, and rank and (ii) situational factors such as kinship and affiliation. When including such covariates in the model, Nakagawa & Schielzeth recommend referring to ‘relative repeatability’ or ‘consistency repeatability’ (Nakagawa & Schielzeth, 2010, page 937). In our original submission, we followed their recommendations, opting for the term ‘relative repeatability’ [e.g., see 114, 289, 349].

However, based on your comment and ones from the other reviewers, we now recognize the need to report an ICC explicitly. As one of the other reviewers pointed out, the most appropriate place to do so is in conjunction with the correlation, which we now do [141-143; 359-361].

Stability within age groups: Splitting the sample into different age categories (to analyze whether the trait becomes increasingly stable across later age groups) is an interesting idea, and one that we have indeed considered. In fact, a more in-depth analysis of developmental trajectories for post-conflict behavior in chimpanzees is the focus of an upcoming study. This will include tests of individual stability across different age-classes for both consolation *and* reconciliation behavior (and relevant comparisons therein). Therefore, we prefer to retain these analyses for a forthcoming paper. However, thank you for encouraging us to at least introduce this idea in the revised discussion of the current manuscript [see 232-234].

I also had a question about some analyses described on page 15. The authors provided their formula for computing the “relative repeatability” but do not cite the relevant formula number or a page number in Nakagawa and Schielzeth (2010) or whether they used some R package to conduct the analyses. I was thus left wondering about the formula because it appears to be incorrect. What the authors appear to be examining here (it’s a really interesting question) is whether consolation is consistent over different recipients of aggression (a sort of cross-situational stability). If so, then the denominator should be the sum of the variance due to bystander ID + variance of the error term, which would be the bystander*recipient interaction (relevant papers, including with applications to animals, are Shavelson et al. 1989, Shavelson and Webb 1991, Figueredo et al. 1995, Hernández-Lloreda and Colmenares 2006). I would like to add to this that, if possible, it would be extremely interesting to examine whether there is consistency in consolation behavior across different aggressors.

You are correct that our models show that consolation is consistent over different recipients of aggression, and thus they provide evidence for a type of cross-situational stability. Thank you for your enthusiasm for this result! Following your encouragement, we have now brought this aspect of our results into greater relief [176-178].

A similarly interesting question is whether such consistency exists across different victims (of which we did find a suggestion—see bottom of Table 1 [136-137]). We have also considered examining cross-situational stability in the latency to console, or the particular behaviors used to console. As these questions are not directly relevant to the question of an *individual’s* consolation tendency, we believe they are beyond the scope of the present research. We do hope to be able to follow up on these important questions by incorporating them into a future paper (see above).

Thank you also for drawing our attention to the need to provide more specific details regarding the source of our statistical equations. In the paper [290], we have now clarified that the equation in question comes from Table 1 in Nakagawa and Schielzeth (2010),

row 4 (repeatability on a latent scale) column 2 (binomial model with a logit link). According to this equation, the repeatability in a multilevel logistic regression is calculated by dividing the individual variance by the total variance + $(\pi^2)/3$ (assuming a dispersion parameter of 1). In the total variance parameter in the denominator, we included (bystander variance + recipient variance) rather than (bystander variance + recipient variance + bystander*recipient variance) because the latter relies on multiple observations of the bystander-recipient pair. After attempting this model, we found that our data did not have enough observations of the bystander-recipient pairs to allow for a reliable estimate of the random effect of pair. As such, we included only bystander and recipient as independent effects.

While we agree that it would be fascinating to discover whether over-and-above individual tendencies, pairs also show unique differences in their tendency to empathize, the present data are not suitable to addressing this question. To do so would have required many observations of every pair of individuals so that it would be possible to test whether a unique pattern of consolation emerged at the pair level. Given the relative infrequency of these events, however, we suspect that the pair-level effect is unlikely to be biologically relevant.

Another problem I identified is that, when reporting p-values, the authors should provide more detail throughout. A good guide that I like to follow comes from some medical journals, e.g., *Psychosomatic Medicine*: “If $p < .10$, then it should be expressed to 3 digits after the decimal point. If the value is $.10$ through $.99$, then it should be expressed to 2 digits. Values of $.000$ and 1.0 should be reported as $< .001$ and $> .99$, respectively.” Moreover, the authors report a $p=0.103$ as indicating that infants are “marginally more likely to provide consolation than juveniles”. It is now recognized that, as p-values are uniformly distributed under the null hypothesis, talk of effects being “trends”, “borderline significant”, etc. is inappropriate (for additional problems with this practice see Wood et al. 2014).

We have followed these guidelines and reworded our statement on the “marginal” p-value [122-125]—thanks for the tip.

A final note with respect to the methods and results: there is some detail missing. For example, concerning multiple comparisons (p. 6) the authors need to note whether these comparisons corrected for multiple tests and, if so, what type of correction was applied. Also, were any variables standardized? If so, the authors should note this and be cautious about whether a b or beta is reported as the latter is a standardized coefficient and the former is not.

Multiple Comparisons: The notion of corrections for multiple comparisons is a controversial and contentious issue. We thank the reviewer for providing us with the opportunity to voice our perspective and pointing out the need to allay future readers’ concerns by addressing them directly in the manuscript. We have now indicated our approach and rationale [346-347].

Along with many biostatisticians, we do not believe that p-correction is the proper solution to multiple testing. Biostatisticians have long argued against the use of p-corrections such as Bonferroni (see, Perneger, 1998; Rothman, 1990; Savitz & Olshan, 1998) and recently these arguments have been gaining the attention of other fields as well (for example, Nakagawa, 2004; Cambiaghi et al., 2011; Jin, Hu, Mathers, & Agmon, 2003; Schwarz et al., 2011; Stein, Fallin, Schork, & Gelernter, 2005). At the heart of the issue is the notion that the interpretation of any single finding should depend on the number of overall tests performed. This logic is problematical for several reasons. First, it discourages thorough data exploration. Second, it increases the chances of type II errors, i.e., missing real patterns. Moreover, it assumes no effect for all relationships and is thus not appropriate when controlling for known effects, e.g., the influence of relationship quality on consolation (e.g., Romero & de Waal, 2010; Clay & de Waal, 2013). As such, we tend to agree with these authors and statisticians that in cases such as ours (controlling for previously established patterns and exploring new, specifically predicted effects) corrections for multiple testing are not warranted. The reviewer's comment has, however, made us realize that we should use this opportunity to weigh in on this debate, which we now do [346-347].

Lastly, we have clarified in the revision that our variables were unstandardized (reporting b rather than β).

3. With respect to their Discussion, the authors claim that the development of empathy in chimpanzees is the opposite to what it is in humans. I do not believe that their data shows this nor do I believe that this difference could be attributable to the use of self-report measures in many human studies. Starting with the latter claim, studies that use peer-ratings find evidence for the same developmental differences in many cultures (McCrae et al. 2005), studies of chimpanzee ratings find that Agreeableness increases, and finding in humans that use behavioral observations also report that older people are higher in empathy (Richter and Kunzmann 2011). What I suspect is going on is that the apparent difference in human and chimpanzee age trajectories can be explained by the well-known problem of identifying what animal personality trait or traits are measured by one or more behavioral measures (Carter et al. 2012a). For example, two seemingly similar measures of behavior in chacma baboons measure two different traits (Carter et al. 2012b). As such, I suspect that consolation behavior is a manifestation of high Agreeableness and high Extraversion, i.e., it is a measure of tendency to feel empathy and the tendency to interact with conspecifics or act on those feelings. Because in chimpanzees, Extraversion declines are much larger than Agreeableness increases, any measure that assesses both domains would show a decline. Evidence for this explanation is in the present results: although human women are higher in Extraversion than men, the reverse is true in chimpanzees (e.g., King et al. 2008). With this in mind I checked the direction of sex differences for consolation behavior. As I predicted, female chimpanzees were lower than males (see Table 1). To address this, the authors should raise the above explanation as a possibility and drop the discussion about self-reports as a possible explanation, which, frankly, adds little to the paper as many of these concerns about self-reports have long been shown to be groundless. I would also like to

encourage them to come up with a way to test this possibility and to think about what it would mean if the developmental trends are similar.

We appreciate these points and ideas. We certainly do not want to make any speculations that could potentially detract from the primary purpose of our paper—that is, to demonstrate the presence and relative stability of individual differences in consolation behavior.

We have increased the precision of our wording throughout the revision to reflect an important distinction—our findings do not point to a drop in empathy per se [208-211], but demonstrate that consolation *behavior* tends to decrease with age, a result that is consistent with research in other apes (Clay & de Waal, 2014a; 2014b; Cordoni et al., 2004).

The idea that consolation behavior is a manifestation of high Agreeableness and high Extraversion is really an intriguing one, as is whether age-related changes in these domains could contribute to our result regarding the decrease in consolation over time. We have included and elaborated on this point our revision [224-232], and feel it has greatly enriched our overall discussion. As we mention in the paper, another possibility is that individuals become more discriminating in their consolation efforts as they get older—we would be curious to see how the findings of Almeling et al. (2016) jibe with the Extraversion decline explanation.

Overall, consolation may represent one facet of empathy, and the human and nonhuman primate literatures were contrasted mainly to generate predictions. However, subsequent findings did lead us to question some widely held assumptions about its trajectory, with a closer examination of the literature revealing many mixed findings. For example, though Richter & Kunzmann (2001) found that empathy increases with age, they only compared younger to older adults (all > 20 years of age); similarly, McCrae et al.'s (2005) study targets were all college-aged or adults. Even focusing within adulthood, some studies point to age-related stability in empathy (Diehl, Coyle, & Labouvie-Vief, 1996; Eysenck, Pearson, Easting, & Allsopp, 1985), whereas others point to a pattern of negative age differences (Grühn, Rebucal, Diehl, Lumley, & Labouvie-Vief, 2008; Helson, Jones, & Kwan, 2002; Phillips, MacLean, & Allen, 2002; Schieman & van Gundy, 2000). This all invites us to speculate that our current understanding of empathy across the entire age spectrum remains rather limited.

Several other points: We do not want our criticisms of self-report methodologies to detract from our main purposes, and therefore have toned down our arguments here. Thank you for suggesting that we do so. On the other hand, we respectfully disagree that such criticisms are groundless; past work has suggested that correlations between self-reported and performance-based empathy measures are small and often non-significant (e.g., Anastassiou-Hadjicharalambous & Warden, 2007; Eisenberg & Lennon, 1983; Ickes, Buysse et al., 2000; Ickes, Stinson, Bissonnette, & Garcia, 1990; Levenson & Ruef, 1992; Marangoni, Garcia, Ickes, & Teng, 1995; Michalska, Kinzler, & Decety, 2013).

We now highlight this potential caveat instead of criticizing self-report methodologies in and of themselves [202-206].

Lastly, our apologies, but we are a bit confused about your point on the direction of sex differences (females being lower than males). We did not find any significant sex differences in consolation behavior (this is consistent with the results presented in Table 1). Actually, previous work on a subset of individuals from these same groups has shown the opposite pattern—overall, adult females offer consolation more often than do adult males (Romero et al., 2010). We hope that this has helped to resolve any potential misunderstandings here.

In addition to these more substantive comments, I had several minor suggestions.

Page 2

Please note the sample size, length of follow-up times, and number of observations in the Abstract.

We have now incorporated this information in our abstract [16-17].

I think the phrase “empathetic personalities” is awkward. The phrase “individual differences in empathy” is a bit better.

We have re-worded this sentence so this phrase now reads: “individual differences in consolation behaviour” [18].

I think “empathy driven response” should simply be “traits”.

We have changed this accordingly—thanks [19].

Please note what the “insights” from this study were?

Here the insights refer broadly to the capacity for and ontogeny of empathy across species. We have reworded this slightly to better capture our intended meaning [23].

Page 3

Please change “Not only does it” to “Not only does empathy”. I think the “but” in this sentence can be deleted.

Done! [26-27]

I would delete the word “accumulating”. Also, variations of this sentence are found throughout the animal personality literature. It would be nice to see a new way to start.

In this paragraph we refer to the empathy literature (rather than the animal personality literature), and would like to point out that evidence is still accruing.

I would delete the word “facets” as it is not really the same as “levels” in this context.

We agree that “facets” and “levels” are different, and meant to emphasize both options— However, to avoid any confusion, we have changed it to just “facets” [29].

What are the “empathetic tendencies” that are found early in life?

We have now rephrased this to underscore that we mean the concern for distressed others (i.e., empathetic concern) [36].

I think the word “both” in the center of the page can be deleted.

We prefer to keep this (increase applies to frequency *and* to complexity) [38-39].

The phrase “Despite these changes” is an awkward way to start the sentence. I think the word “also” can be deleted in the same sentence.

We have taken your suggestion to rephrase this sentence [42-43].

There is no need to italicize “individual differences”.

Agreed—we have removed italics [42].

The phrase “conceptualized as part of a broader prosocial personality” is awkward. Do you mean personality dimension or domain?

Thanks for pointing this out, we have edited this statement to be more explicit [46].

Page 4

I would change “remain hard to elucidate” to “are yet to be elucidated”.

Although a number of attempts have been made, uncovering the precise cognitive and emotional capacities required for consolation remains inherently complex. To us, “are yet to be elucidated” seems to imply they have not yet been studied, thus we prefer our original wording.

The reference here to the oxytocin literature should be a bit uncritical, particularly given that the human findings related to oxytocin are wanting (Nave et al. 2015, Lane et al. 2016).

We agree that there are many methodological challenges when studying the neurochemical underpinnings of empathy, and that results should be cautiously interpreted—especially when evaluating studies using intranasal administration of oxytocin or correlational studies. However, the work cited here refers to a pharmacological intervention in rodents (i.e., direct infusion of oxytocin antagonist in specific brain areas), which demonstrates that the activation of oxytocin receptors in the anterior cingulate cortex is necessary for consolation behavior.

I do not think the sentence “However, these studies failed to...” is necessary.

We prefer to keep this in [70-71], as these studies were not explicitly designed to test for stable individual differences or age-related differences (for example, the bonobo studies included few older subjects, whereas the gorilla research included very few younger subjects, making it difficult to draw any formal conclusions about age or individual differences).

Again, the italics here are not needed.

Agreed.

Page 5

I would change “also exist” to “exist”.

We have opted to keep our original wording, as it insinuates that stable individual differences in empathy have been studied in humans [78].

Here and elsewhere, I would change “a number of” to “numerous”, “several”, “many”, or the like.

We have made such changes.

I think the parenthetical material that follows “Finally” is probably better separated off with commas.

Again, we are in agreement—thanks for this suggestion [94-95].

Here again, what “novel insights” are you expecting?

The novel insights here refer to our main findings—namely, that stable individual differences exist in chimpanzee consolation behavior, that infants/juveniles console more often than adolescents/adults, and that consolation predicts social integration.

I would delete “a variety of”.

Deleted.

Here you insert “see” before calling out a table whereas you do not do so elsewhere.

OK—we have edited this to be more consistent throughout.

Page 6

I would delete the word “importantly”.

Deleted.

.052 is quite small. I wonder if this is because of the possible problem with how this was computed (see my earlier comments).

Please see above comment.

Page 7

There should be error bars in the figure. Also, in the table (and in other tables), I would substitute “Test Statistic” with the statistic (z in the present case). The p-values in the table that are below .001 should be “< .001” and not “.000”.

We have omitted error bars because the figure is already (necessarily) border-line cluttered, and we are plotting the model predictions, not the data. At the Editor’s discretion, we would prefer to keep Figure 3 as-is.

In our results table [136-137], the test statistic is different for fixed effects (Z values) and random effects (χ^2 values). We have thus labeled the overall column ‘Test Statistic.’

Page 8

Does “Figure 1” here (first paragraph) actually refer to Figure 2? I could not find any call out for Figure 2 and it seems to be more consistent with the results described here.

Yes—thanks for drawing our attention to this error; we have corrected it [141].

Page 9

Please change “relation” to “relationship”.

We have done so [139].

I do not think that Figure 3 is cited in the text.

Again, thanks for calling this to our attention! It is now cited [160].

Page 10

Please avoid so-called statements of priority. Just because a study is first does not mean it is interesting or important.

In principle we agree, and have thus tried to tone down such statements of priority [168-176].

I would delete “distinct” here.

Deleted.

Page 11

I would delete “and/or temperament” as the distinction between temperament and personality is arbitrary. Please note, too, that environmental factors can also be individual as is the case with the nonshared environment that behavior geneticists find to be important.

Yes, we agree, environmental factors can also be individual. We have now clarified this by referring to ecological (vs. social) factors [187-190].

I think the speculation here goes too far, particularly given the question of whether the differences are, in fact, present (see my earlier comments). However, the point about needing to compare these systems in humans and chimpanzees is an excellent one.

We really appreciate these comments, and have now edited this paragraph to stress the speculative nature of our statement, along with incorporating the valuable comments of another review on this same point [190-195].

Page 12

It is not clear to me what the authors are trying to say in their sentence that begins “We should add...” I suggest revising this to make it clear.

Here we mean that individuals may become more filtered or selective in their expressions of empathy over time. We have amended this section to be more clear [208-211].

In the first full paragraph, what result does “This result” refer to?

This refers to our finding that consolation behavior appears to decline with age, a point which we have now made more explicit in the revision [212-213].

The point made in the sentence “Another possibility is that younger...” is excellent. I recall hearing from a colleague some time ago that there is a human literature on this kind of thing, too, though I cannot remember any specifics. I would thus encourage the authors to see if they can identify at least one relevant study or review in humans.

Yes, there is a human literature on the general idea that as people get older, they become more selective in their social partners/efforts. ‘Socioemotional selectivity theory,’ developed by Laura Carstensen and her colleagues (see ref. 52 in the manuscript for a nice review) describes this phenomenon. To summarize: as humans age (and time is perceived as more and more limited), emotional goals become more salient than knowledge goals. Thus, the desire to affiliate with unfamiliar people (‘expand one’s horizons’ and gain novel information) decreases while the preference for familiarity increases. Thanks for encouraging us to look into this further—we have now made note of this in the paper [222-223]; it is indeed an interesting body of work!

Page 13

The authors here begin talking about “emotional competence” which was not raised earlier. I think they should stick to the terminology they used throughout.

This study found that consolation was predicted by overall social competence and effective emotion regulation (as reflected in the speed of recovery from self-distress and behavioral measures of anxiety). We have now edited this section to be more clear and consistent in our terminology [242-246].

The four parenthetical statements that begin “e.g.” are a distraction. I would revise so as to avoid this.

We have now revised this last sentence to avoid this [246-250].

I think “more detailed” can simply be “detailed”.

Yes agreed [261].

Page 14

I would change “behavioural elements” to “behaviours”.

Done [269].

Page 15

I think the authors can delete “Specifically” in this instance.

Yes agreed [301].

The sentence that begins “Provided the fixed...” is a bit awkward and too long.

We have amended our wording in this section to clarify our statistical rationale.

The sentence that begins “Using this test...” is also too long and it repeats information from earlier.

Here we reemphasize a key point (although we want to avoid redundancies, we also want to be sure our statistical approach is clear).

Supplementary Figure 1

Here it would be worth spelling out what FS1 and FS2 mean. I know it is in the text, but it should be in the figure legend.

We have now added this information in Figure S1’s legend.

Thank you for asking me to review this interesting manuscript. I hope the authors find my comments helpful.

Very helpful indeed! Thank you again. All references mentioned above that are not included in our revised manuscript are appended below.

Alexander Weiss [I sign my reviews]

References (that are not already cited in manuscript):

Allegue, H., Araya-Ajoy, Y. G., Dingemanse, N. J., Dochtermann, N. A., Garamszegi, L. Z., Nakagawa, S., ... Westneat, D. F. Statistical Quantification of Individual Differences (SQuID): An educational and statistical tool for understanding multilevel phenotypic data in linear mixed models. *Methods in Ecology and Evolution* (2016).

Cambiaghi, M., Teneud, L., Velikova, S., Gonzalez-Rosa, J. J., Cursi, M., Comi, G., & Leocani, L. Flash visual evoked potentials in mice can be modulated by transcranial direct current stimulation. *Neuroscience* **185**, 161–165 (2011).

Diehl, M., Coyle, N., & Labouvie-Vief, G. Age and sex differences in strategies of coping and defense across the life span. *Psychol. Aging* **11**, 127–139 (1996).

Eysenck, S. B., Pearson, P. R., Easting, G., & Allsopp, J. Age norms for impulsiveness, venturesomeness and empathy in adults. *Pers. Individ. Dif.* **6**, 613–619 (1985).

Ickes, W., Buysse, A., Pham, H., Rivers, K., Erickson, J. R., Hancock, M., ... Gesn, P. R. On the difficulty of distinguishing “good” and “poor” perceivers: A social relations analysis of empathic accuracy data. *Pers Relatsh* **7**, 219–234 (2000).

Jin, X., Hu, H., Mathers, P. H., & Agmon, A. Brain-derived neurotrophic factor mediates activity-dependent dendritic growth in nonpyramidal neocortical interneurons in developing organotypic cultures. *J. Neurosci* **23**, 5662–5673 (2003).

Levenson, R. W., & Ruef, A. M. Empathy: A physiological substrate. *J. Pers. Soc. Psychol.* **63** 234–246 (1992).

Marangoni, C., Garcia, S., Ickes, W., & Teng, G. Empathic accuracy in a clinically relevant setting. *J. Pers. Soc. Psychol.* **68** 854–869 (1995).

McCrae R. R., Terracciano, A., & 78 Members of the Personality Profiles of Cultures Project. Universal features of personality traits from the observer's perspective: Data from 50 cultures. *J. Pers. Soc. Psychol.* **88**, 547–561 (2005).

Michalska, K. J., Kinzler, K. D., & Decety, J. Age-related sex differences in explicit measures of empathy do not predict brain responses across childhood and adolescence. *Dev. Cogn. Neurosci.* **3**, 22–32 (2013).

Perneger, T. V. What's wrong with Bonferroni adjustments. *Brit. Med. J.* **316**, 1236–1238 (1998).

Phillips, L. H., MacLean, R. D., & Allen, R. Age and the understanding of emotions: Neuropsychological and sociocognitive perspectives. *J. Gerontol. B. Psychol. Sci. Soc. Sci.* **57**, 526–530 (2002).

Rothman, K. J. No adjustments are needed for multiple comparisons. *Epidemiology* **1**, 43–46 (1990).

Savitz, D. A., & Olshan, A. F. Describing data requires no adjustment for multiple comparisons: A reply from Savitz and Olshan. *Am. J. Epidemiol* **147**, 813–814 (1998).

Schieman, S., & van Gundy, K. The personal and social links between age and self-reported empathy. *Soc. Psychol. Quart.* **63**, 152–174 (2000).

Schwarz, E., Guest, P. C., Rahmoune, H., Wang, L., Levin, Y., Ingudomnukul, E., ...Bahn, S. Sex-specific serum biomarker patterns in adults with Asperger's syndrome. *Mol. Psychiatry* **16**, 1213–1220 (2011).

Stein, M. B., Fallin, M. D., Schork, N. J., & Gelernter, J. (2005). COMT polymorphisms and anxiety-related personality traits. *Neuropsychopharmacology* **30**, 2092–2102 (2005).

References

Ashton MC, Lee K (2007) Empirical, theoretical, and practical advantages of the HEXACO model of personality structure. *Personality and Social Psychology Review* 11:150-166

Bell AM, Hankison SJ, Laskowski KL (2009) The repeatability of behaviour: a meta-analysis. *Anim Behav* 77:771-783

Carter AJ, Feeney WE, Marshall HH, et al. (2012a) Animal personality: What are behavioural ecologists measuring? *Biological Reviews* 88:465-475

Carter AJ, Marshall HH, Heinsohn R, et al. (2012b) How not to measure boldness: Demonstration of a jingle fallacy in a wild social primate. *Anim Behav* 84:603-609

Costa PT, Jr., McCrae RR (1995) Domains and facets: Hierarchical personality assessment using the Revised NEO Personality Inventory. *J Pers Assess* 64:21-50

Figueredo AJ, Cox RL, Rhine RJ (1995) A generalizability analysis of subjective personality assessments in the stump-tail macaque and the zebra finch. *Multivar Behav Res* 30:167-197

Freeman HD, Brosnan SF, Hopper LM, et al. (2013) Developing a comprehensive and comparative questionnaire for measuring personality in chimpanzees using a simultaneous top-down/bottom-up design. *Am J Primatol* 75:1042-1053

Hernández-Lloreda MV, Colmenares F (2006) The utility of generalizability theory in the study of animal behaviour. *Anim Behav* 71:983-988

King JE, Figueredo AJ (1997) The Five-Factor Model plus Dominance in chimpanzee personality. *J Res Pers* 31:257-271

King JE, Weiss A, Farmer KH (2005) A chimpanzee (*Pan troglodytes*) analogue of cross-national generalization of personality structure: Zoological parks and an African sanctuary. *J Pers* 73:389-410

King JE, Weiss A, Sisco MM (2008) Aping humans: Age and sex effects in chimpanzee (*Pan troglodytes*) and human (*Homo sapiens*) personality. *J Comp Psychol* 122:418-427

Lane A, Luminet O, Nave G, et al. (2016) Is there a publication bias in behavioural intranasal oxytocin research on humans? Opening the file drawer of one laboratory. *J Neuroendocrinol* 28

McCrae RR, Costa PT, Jr. (1991) Adding Liebe und Arbeit: The full Five-Factor Model and well-being. *Pers Soc Psychol Bull* 17:227-232

McCrae RR, Terracciano A, 78 Members of the Personality Profiles of Cultures Project (2005) Universal features of personality traits from the observer's perspective: Data from 50 cultures. *J Pers Soc Psychol* 88:547-561

Nakagawa S, Schielzeth H (2010) Repeatability for Gaussian and non-Gaussian data: A practical guide for biologists. *Biological Reviews* 85:935-956

Nave G, Camerer C, McCullough M (2015) Does oxytocin increase trust in humans? A critical review of research. *Persp Psychol Sci* 10:772-789

Richter D, Kunzmann U (2011) Age differences in three facets of empathy: Performance-based evidence. *Psychol Aging* 26:60-70

Roberts BW, DelVecchio WF (2000) The rank-order consistency of personality traits from childhood to old age: A quantitative review of longitudinal studies. *Psychol Bull* 126:3-25

Roberts BW, Walton KE, Viechtbauer W (2006) Patterns of mean-level change in personality traits across the life course: A meta-analysis of longitudinal studies. *Psychol Bull* 132:1-25

Shavelson RJ, Webb NM (1991) *Generalizability theory: A primer*. Sage, Thousand Oaks, CA

Shavelson RJ, Webb NM, Rowley GL (1989) Generalizability theory. *Am Psychol* 44:922-932

Terracciano A, McCrae RR, Brant LJ, et al. (2005) Hierarchical linear modeling analyses of the NEO-PI-R scales in the Baltimore Longitudinal Study of Aging. *Psychol Aging* 20:493-506

Weiss A, Inoue-Murayama M, King JE, et al. (2012) All too human? Chimpanzee and orang-utan personalities are not anthropomorphic projections. *Anim Behav* 83:1355-1365

Weiss A, King JE, Hopkins WD (2007) A cross-setting study of chimpanzee (*Pan troglodytes*) personality structure and development: Zoological parks and Yerkes National Primate Research Center. *Am J Primatol* 69:1264-1277

Wood J, Freemantle N, King M, et al. (2014) Trap of trends to statistical significance: likelihood of near significant P value becoming more significant with extra data. *Br Med J* 348:g2215

Reviewer #2 (Remarks to the Author):

The authors report an interesting study on consistent variation regarding consolation in chimpanzees. They analyzed a very extensive long-term (up to 8 year) behavioural dataset of a total of 44 chimpanzees and found that the tendency to console others after a conflict varies significantly among those individuals, and that this variation is consistent over a long period of time / age-classes. Moreover, they show that even though this variation remains consistent, the general tendency to console decreases with age in chimpanzees. I think this is an interesting, well performed study. However, I have one major conceptual concern with the study that puts the novelty of this study into question, and therefore, I feel this study doesn't warrant publication in *Nature*

Communications and would better fit a more topical journal.

Thank you for these suggestions; please find our detailed responses below.

Major concern:

The authors claim that they have found empathic personalities, which is based on the premise that consolation has empathic motivations. However, this does not necessarily need to be the case in chimpanzees, as has been nicely reviewed by a paper by Fraser and colleagues (2009 *Communicative & Integrative Biology*), which the authors do not cite. Some studies have shown that what is generally referred to as post-conflict 3-rd party affiliation, can also be a way to avoid redirected aggression. As such, rank effects can also play an important effect in for example the opportunity to console, which is an important factor in calculating the variation in consolation in this manuscript.

Given that I'm not convinced that the behaviour the authors report here equals empathy, we're left with a study that shows long-term consistency of a behaviour; i.e. personality, which is not very novel.

We agree that what is generally referred to as post-conflict third party affiliation is a heterogeneous phenomenon whose function varies depending on the species and the social context in which it occurs. In our study, however, consolation is defined as the first affiliative behavior directed from bystanders to recent recipients of aggression. This definition excludes other types of post-conflict third-party affiliation, such third-party contacts sought by the conflict participants themselves or made with the aggressor [54-55].

We argue that consolation in chimpanzees and other great apes most likely rests on empathy because findings from several independent studies have shown that: a) its main function is to reduce the victim's distress, b) alternative hypotheses have been rejected (see below), and c) it fits all predictions derived from the empathy-based theory (Kutsukake & Castles, 2004; Palagi et al., 2004; Palagi et al., 2006; Fraser et al., 2008; Romero et al., 2010; Romero & de Waal, 2010; Clay & de Waal, 2013a, 2013b; Palagi & Norscia, 2013, see Fraser & Bugnyar, 2010; Palagi et al., 2014; Burkett et al., 2016, for similar results in other species).

Furthermore, detailed functional analyses on consolation behavior in great apes have failed to find support for the self-protection hypothesis mentioned by the reviewer (Wittig & Boesch, 2010; Romero et al., 2010; Romero & de Waal, 2010; Clay & de Waal, 2013; Palagi & Norscia, 2013), with one exception (Koski & Sterck, 2009). It is worth noting that study subjects from this last study experienced considerably higher risk of receiving redirected aggression during the post-conflict period (i.e., 10.8% of post-conflict periods; Koski & Sterck, 2009) than chimpanzees living in groups where the use of consolation as self-protection seems to be absent (< 0.5% of post-conflict periods in our study groups, Romero & de Waal, 2010). Moreover, in our study groups, recipients of aggression do

not receive post-conflict friendly contacts from their typical targets of redirected aggression, but from their close associates (i.e., kin and friends).

Fraser and colleagues' (2009) work, which is now cited in our manuscript [59 and 280-282], concluded that the function of these post-conflict affiliations—and hence their underlying mechanisms—is directly related to the quality of the relationships between the individuals involved, with contacts between close associates likely functioning as consolation. Thus, the consolation explanation, rather than the self-protective hypothesis, applies to our observations.

In our multivariate analyses, we control not only for rank effects—as the reviewer suggested—but also for all other variables known to affect the occurrence of consolation in chimpanzees (Romero et al., 2010; Fraser et al., 2008; Palagi et al., 2006). We would like to direct the reviewer to our Methods [328-338] for a full description of the variables included in our analysis.

Although I applaud the authors for taking the observational approach, it should also be noted that personality studies on chimpanzees that use the questionnaire approach use 'Conciliatory' as one of the adjectives describing their subjects (see for the original adjective list Goldberg 1990 JPSP) and also show consistency in its overarching construct (personality factor) Agreeableness in chimpanzees, and some studies also report long-term consistency (King et al. 2008; JCP). Also that latter study reports aging effects on personality traits and actually reports an opposite effect; i.e. Agreeableness increase with age, albeit only slightly. Again this paper is not, yet should be cited.

The reviewer is absolutely correct—this literature is highly relevant and should be addressed in our paper. We have now incorporated several studies on Agreeableness in chimpanzees [83-85], including King and colleagues' (2008) work revealing age-related increases in this personality domain. We also took the suggestion of other reviewers to reference the human research on Agreeableness in the introduction [79-83]. Although 'conciliatory' was indeed on Goldberg's (1990) original adjective list, the chimpanzee studies following the rating-based method used other descriptors for Agreeableness: 'sensitive,' 'gentle,' 'protective,' 'sympathetic,' and 'helpful' (e.g., King et al., 2005; 2008; Weiss et al., 2000; 2007; 2012).

Intriguingly, while these are certainly relevant adjectives to consolation behavior, another reviewer also pointed out that conciliatory tendencies may tap into the Extraversion personality domain as well. It is possible, then, that consolation is a manifestation of both Agreeableness (sensitivity towards others' needs) and Extraversion (tendency to seek out and actively engage with others). We have addressed this interesting possibility in our discussion [224-232].

Lastly, thank you for applauding our use of the observational approach—in the revised manuscript, we have taken the opportunity to highlight our use of a behavioral measure

of empathy [85-87].

Minor comments:

throughout: don't say consolation is empathy driven and actually refer to it as post-conflict 3-rd party affiliation. I'm fine with speculation, yet don't take it as a truth.

The term post-conflict third party affiliation refers to any interaction between one of the opponents and a third-party during the post-conflict period. Because we investigate only affiliative interactions directed from uninvolved bystanders toward recipients of aggression [54-55], and because these interactions are most likely to rest on empathy (please, see comment above), we have opted to keep the term consolation.

I. 47. ALL animals??

This now reads “many animals” [51]. Thanks.

I. 58. please provide refs

We have now provided supporting references here [65-67].

I. 61. ref 20, 23. You should be cautious with referring to the age effects in these studies since the sample of Bonobos in these studies contained very few older individuals (i.e. only 8 older than or equal to 15, and only 2 older than or equal to 20).

The reviewer is correct in noting that the bonobo research cited here included very few older individuals (interestingly, the gorilla research, also cited here, included very few younger individuals). For this reason, we then go on to advise that the results of these studies be interpreted with caution: “However, these studies failed to examine if the reported differences reflect age characteristics or stable individual differences, or both. More broadly, insufficient sample sizes and limited longitudinal data have precluded formal conclusions regarding developmental questions” [70-72].

I. 105-112. Did you perform any correction for multiple comparisons?

We ran these comparisons without adjustment for multiple hypothesis testing following the recommendation of (Nakagawa, 2004; Perneger, 1998; Rothman, 1990; Savitz & Olshan, 1998). We have now added this to our Methods [346-347].

I. 120. Please mention in the table what where the reference categories.

The reference categories are included in Table 2 [313-314; i.e., where we describe the variables included in Table 1], and denoted by superscript ¹.

I. 122. To calculate repeatability, ICC's are more appropriate as they specifically take inter-individual variation into account (see Lessels & Boag 1987 AUK)

We apologize for our lack of clarity regarding our statistical reporting. Following Nakagawa & Schielzeth (2010), by reporting relative repeatability, we did indeed calculate an equivalent to the ICC. From your comment and similar comments from other reviewers, we have realized the need to explicitly report ICCs in addition to our other statistics, which we have now done [143-144; 359-361].

I. 253. Please mention how age-classes are defined

Please refer to our Methods for information on how age-classes were defined [339-341]—i.e., “Age-classes were defined as follows: infants (1–4 years old), juveniles (5–7 years old), adolescents (8–9 years old), and adults (10 years and above).”

I. 275. What were the reference groups for Affiliation, Kinship, Sex, and recipient sex?

We dummy coded these variables so that the coefficient showed the effect of the variable name—e.g., the kinship coefficient estimated the effect of the pair being kin vs. non-kin. As such, the reference group was the group coded as zero (see Table 2 [313-314]).

I. 294. why did the authors choose a binary way of analysing affiliative relationships rather than putting it in as a continuous variable, if you have the data anyway?

Using the continuous variable instead of the binary one might have been more informative if the objective of our study were to explore the social determinants of consolation behavior. However, our interest was not to evaluate the effect of affiliation level on the occurrence of consolation, but to control for the known effect of social closeness on consolation (e.g., Palagi et al., 2004; Palagi et al., 2006; Fraser et al., 2008; Romero et al., 2010; Clay & de Waal, 2013; Palagi et al., 2014). Therefore, for each individual we identified its closest associates and classified the rest of its group-mates as not-close associates.

I. 312. What is an age-period. I think this part needs a bit more explanation.

Age period refers to the entire timeframe in which a subject fell within a particular age-class (as defined on [339-341]). Consolation was then calculated per each age period (or periods) to determine whether individuals’ tendencies when young correlated with their tendencies when old. We have now clarified what an age-period is in the revision [352-353].

References (that are not already cited in manuscript):

Fraser, O. N., & Bugnyar, T. Do ravens show consolation? Responses to distressed others. *PLoS ONE* **5**, e10605 (2010).

King, J. E., Weiss, A., & Farmer, K. H. A chimpanzee (*Pan troglodytes*) analogue of cross-national generalization of personality structure: Zoological parks and an African sanctuary. *J. Pers.* **73**, 389–410 (2005).

- Koski, S. E., & Sterck, E. H. M. Post-conflict third-party affiliation in chimpanzees: What's in it for the third party? *Am. J. Primatol.* **71**, 409–418 (2009).
- Kutsukake, N., & Castles, D. L. Reconciliation and post-conflict third-party affiliation among wild chimpanzees in the Mahale Mountains, Tanzania. *Primates* **45**, 157–165 (2004).
- Palagi, E., Cordoni, G., & Tarli, S. B. Possible roles of consolation in captive chimpanzees (*Pan troglodytes*). *Am. J. Phys. Anthropol.* **129**, 105–111 (2006).
- Palagi, E., & Norscia, I. Bonobos protect and console friends and kin. *PLoS ONE* **8**, e79230 (2013).
- Palagi, E., Paoli, T., & Tarli, S. B. Reconciliation and consolation in captive bonobos (*Pan paniscus*). *Am. J. Primatol.* **62**, 15–30 (2004).
- Perneger, T. V. What's wrong with Bonferroni adjustments. *Brit. Med. J.* **316**, 1236–1238 (1998).
- Rothman, K. J. No adjustments are needed for multiple comparisons. *Epidemiology* **1**, 43–46 (1990).
- Savitz, D. A., & Olshan, A. F. Describing data requires no adjustment for multiple comparisons: A reply from Savitz and Olshan. *Am. J. Epidemiol* **147**, 813–814 (1998).
- Weiss, A., King, J. E., Figueredo, A. J. The heritability of personality factors in chimpanzees (*Pan troglodytes*). *Behav. Genet.* **30**, 213–221 (2010).
- Weiss, A., Inoue-Murayama, M., King, J. E., Adams, M. J., & Matsuzawa, T. All too human? Chimpanzee and orangutan personalities are not anthropomorphic projections. *Anim. Behav.* **83**, 1355–1365 (2012).
- Wittig, R. M., & Boesch, C. Receiving post-conflict affiliation from the enemy's friend reconciles former opponents. *PLoS ONE* **15**, e13995 (2010).

Reviewer #3 (Remarks to the Author):

The paper reports individual stability in the tendency to offer post-conflict consolation in captive chimpanzees. The study takes advantage of the large data set on post-conflict behaviour in the records of the Yerkes chimpanzees. The aim taps nicely into the hot topic of behavioural consistency vs. plasticity (i.e. personality) and assesses it in the realm of social behaviour. The results are interesting, given that we still know little of the mechanisms of consolation and because personality in social behaviour is still relatively poorly studied. The study is likely to prompt further research into the personality effects on social behaviour, and into the factors influencing socio-positive and prosocial

behaviour.

However, I have some issues with the paper with regards to the methods and the scope.

Thank you for your interest and feedback. Please find our responses to your comments below.

One of my concerns is about the way the study assesses consolation. As we know by now, not all bystander-initiated affiliative contacts to a conflict victim necessarily serve the consolatory function. It is commonplace to lump them, but it turns out that there are several functions to the seemingly similar-looking behaviour (Fraser et al. 2009). I do realise that in this kind of a data set it is not possible to pull these functionally different behaviours apart. However, I would appreciate at least an acknowledgment of this caveat in the introduction.

We agree that it is possible that some of the observed affiliations might be functionally different than consolation. However, results from previous analyses suggest that these post-conflict affiliative contacts mainly function as consolation in our study groups (see Romero & de Waal, 2010; Romero et al., 2010). Nonetheless, following your suggestion, we have acknowledged this possibility in the manuscript [59 and 280-282].

Secondly, the data were never collected to study post-conflict behaviour. Therefore, a major caveat is that there is no way of controlling the effect of individuals' baseline affiliation to partial it out from 'true' consolation. In controlled studies, this is done with well-established methods that yield a 'corrected consolation tendency' for each individual. This is not possible to derive for this study. Therefore, there is inevitably the possibility that individuals' post-conflict affiliation tendency is influenced by their general affiliation tendency. However, the group has published a number of papers on consolation with this dataset and acknowledged this caveat in their earlier papers. A note on this would be needed here, too, at least as a discussion point. Both of these issues are, in my opinion, highly relevant in the discourse of consolation behaviour, but given the weight of the group's earlier papers with this same dataset, I suppose we just have to agree to disagree, and move on.

Although the observation sessions were not specifically designed to study post-conflict behavior, previous work using the same dataset has indeed used the well-established PC-MC method to study consolation (for details, see pg. 280, Romero & de Waal, 2010). This includes multivariate analyses using the aforementioned corrected consolation tendency (pg. 281-282, Table 4, Romero & de Waal, 2010), whose results show that the observed pattern of consolation is not just a mirror of an individual's general affiliative pattern. Since these results have already been published, we did not run similar analyses in the present study.

That said, the following issues should be taken into account in the revision.

- The paper is framed in line with personality literature and the title suggests empathic tendency as a personality trait. Yet, the background of and implications on personality

literature are very thin. As links are drawn to the stability of human empathic tendency, the literature on human personality is highly relevant, but totally absent in the paper. Similarly, there is no introduction into animal personality in general or chimpanzee personality in particular. Since there is a large body of research on primate personality available by now, this needs to be tied into the paper.

We very much agree here. Thanks for this suggestion (your concern was also shared by both other reviewers). Accordingly, we have included the human personality literature in our revised introduction [79-87] and discussion [223-234]. In those sections we have also incorporated more work on chimpanzee personality in general, particularly with respect to how consolation and empathy might fit under the broader personality frameworks elucidated by commonly used psychometric models. For example, as detailed below, our revision includes the research on Agreeableness in both humans and other primates, which is highly pertinent to our work. We hope our new approach brings both more breadth and cohesion to our work by drawing interesting and important links between these different areas of personality research.

- The crucial metric to establish 'personality', or relative within-individual consistency in behaviour, is repeatability. In this study repeatability is calculated differently from the usually recommended way in the personality literature. It is relevant to establish not only the amount of variance in one individual's repeated observations but also, in relation to that, the variance among individuals. As far as I understand, the formula on l. 252 does not give that. Please calculate in addition repeatability following Lessells & Boag 1987. It gives a value between 0 and 1, with higher repeatability indicating small contribution of within-individual variance to the overall variance and, thus, allows one to estimate in a direct and intuitively understandable way the degree of within-individual consistency and among-individual differences.

Thank you for these suggestions regarding the best way to communicate our effects. We completely agree on the importance of following field standards for data analysis practices, which is why we used the Nakagawa and Schielzeth (2010) as our primary data analysis guide. This paper currently has 538 citations in Web of Science and has been deemed a "Highly Cited Paper." In it, the authors write, "R [repeatability] is often called the intra-class correlation coefficient (ICC)" (Nakagawa and Schielzeth, 2010; page 937). Furthermore, for non-Gaussian data such as ours, Generalized Multilevel Models do not provide a comparable estimate of between-individual variance (which would be the equivalent of unexplained-variation). Thus, when we reported repeatabilities in our manuscript using this equation (the origin of which we now specify [290]), we were in fact reporting a type of ICC from a recommended source. However, based on your comment and ones from other reviewers, we have now added an ICC statistic to our results section [143-144; see also 359-361 in Methods]. Thank you again for suggesting this.

Specific comments

Introduction

- I. 31: The references here include papers on human empathy, mice prosocial behaviour, and three review papers. As far as I know, these papers don't report the facilitating effect of similarity on empathy in animals. The social closeness effect is shown in primates and the familiarity effect is shown in mice.

We appreciate this comment. We have now included references showing the facilitating effect of similarity on empathy in animals [33].

- I. 41-43. Given that your paper is framed in line with personality literature and the title suggests empathic tendency as a personality trait, the literature on human personality and animal personality are both highly relevant. The studies cited here show nicely the developmental aspects of empathy and prosociality, but why is literature on the association of empathy to human personality structure and the developmental effects thereof ignored? Empathy and prosociality are associated with Agreeableness, one of the five constructs in human personality according to the most widely used psychometric model (the Five Factor Model). Agreeableness shows age-related increase in humans (e.g. McAdams & Olson 2010), and there is a study reporting the same in chimpanzees (albeit with a human-psychology derived method of assessing personality; King et al 2008). This discussion is very relevant to your paper, since you want to establish empathic tendency as a personality trait.

These comments resonate with those of two other reviewers. Although we did not want our introduction's focus to be too broad, we have now included the pertinent literature on human and animal personality [79-87].

Specifically, we add that empathy is commonly associated with the Agreeableness domain described by widely used psychometric personality models. In humans, we describe that longitudinal work has revealed that facets of Agreeableness and related traits show both relative consistency and age-related increases. We also highlight evidence for the stability of Agreeableness in chimpanzees, and for it to be higher among older individuals compared to younger individuals. Importantly, these studies were based on observer ratings (e.g., experimenters coding for descriptor adjectives for each personality domain), rather than a behavioral measure. Therefore, we also take the opportunity to point out this unique contribution of our study. Overall, we appreciate very much your suggestion to include this literature, and concur that it was a necessary addition to our manuscript.

- I. 45-46. Related to the above comment, that is not really true when you consider the human personality research in general. It's also relevant to clarify here what is meant by consistency: a behavioural tendency that does not change or tendency that allows change but shows (relatively high) rank-order stability.

Yes we agree—we meant rank-order stability, which is why we chose to use the term relative stability or relative consistency (rather than absolute). That is, a tendency that is stable relative to other group members. However, to avoid such confusion, we have now

explained this in the introduction [43-45]. Thank you for encouraging us to make this important point of clarification.

We also agree that longitudinal work on Agreeableness has helped to better illuminate the trajectory of this domain—but would respectfully submit that the focus has typically been on older age-classes (e.g., comparing adolescence to adulthood) whereas younger age-classes (i.e., infancy and early childhood) are rarely covered in this literature.

-l. 53ff. As you note, in addition to oxytocin, HPA-axis is known to be involved. This was nicely demonstrated in a recent study on rats: anxiolytic drug blocked helping behaviour that was likely be empathy-driven. Worth citing here? (Ben-Ami Bartal et al 2016)

Thank you for this suggestion. We have now included this work [62].

Methods:

- l. 255 “metric base of repeatability”. This does not really give a metric that would be comparable to repeatability in animal personality studies.

Please see above comment.

- l. 328-320 and l. 234 ff: I don't find this in Table 1. Based on the model description earlier, I thought that Table 1 shows the occurrence of consolation as the response variable and has dyad as a unit of analysis. Where is the model for individual's given consolation in an older age class as the response variable? The results section on this is equally unclear. Given the centrality of this result to your argument, this needs to be clarified.

Please refer to our Methods [356-364]. In this model, an individual's consolation tendency from the younger age period(s) was entered as a predictor of the probability of consolation occurring within the older age period(s) (i.e., the probability of consolation's occurrence, not consolation tendency, was the response variable).

- The approach specified on l. 260 ff is appropriate to establish the amount of variance that is explained by among-individual variation, so no issues there. However, I still argue for the value of establishing the classic sense repeatability values for individuals' consolatory tendency, because as a generally used metric it allows an easy and intuitive way of assessing the relative consistency of behaviour.

Thank you for helping us to ensure that we report intuitive ways of assessing the consistency of consolation behavior. We fully agree on the importance of providing readers with straightforward metrics of behavioral consistency, which was our original rationale for including the correlational analysis in addition to the more complex (and flexible) Generalized Multilevel Model. However, to reach the broadest audience possible, we now see the need to report ICCs as well. We have thus added an ICC analysis to the correlation result [see 143-144; 359-361].

Results

- l. 97: this perfectly illustrates the difficulty with using only a mixed model with other predictors as the only metric for repeatability. $r=0.052$ is a really low figure, but it obviously is directly influenced by the model specifics. So pls see my above comment on calculating repeatability (e.g. with intra-class correlation coefficient, see e.g. McGraw & Wong, 1996).

Thank you for pointing out how the low repeatability is influenced by our modeling specifics. Moreover, the low base rate of consolation makes finding high repeatabilities very difficult with this particular modeling approach. Nevertheless, we feel it is necessary to include this model in the manuscript as it is the only way to control for potential confounding individual-level effects such as sex and situational factors such as kinship. Without controlling for such effects, it would be possible that our results were not driven by individual differences in consolation, but simply consistency by sex, rank, kinship etc.

However, as we mention above, following your suggestion (and those from other reviewers), we now include an ICC based on the correlation data [see 143-144; 359-361].

- l. 105ff: The betas of adolescents and adults here are missing the negative sign (provided they are correct in the Table 1). The text is also somewhat confusing in wording, it's not always clear which group the stats in parentheses refer to.

Thank you—negative signs have been added to correspond correctly to our results/Table 1. Further, we have edited our wording to make these comparisons more straightforward [122-129].

- l. 122 ff: Not Fig 1 but Fig 2, I suppose.

Yes, we meant Figure 2, thank you for drawing our attention to this [141].

- l. 140 ff: Why did you not run a model of this one? I would think that individual's CSI is also influenced by rank and sex, so having CSI as a response variable and sex, rank, age, and consolation tendency as predictors would nicely show its relative predicting value.

We have now entered CSI in the full model (as a predictor of consolation), and presented this result [162-163].

Discussion

- l. 158-159: here a broader incorporation into the literature of animal personality is pertinent. How does consistency in this form of social behaviour potentially relate to other findings on chimpanzee (social) personality traits? What would you predict for other social species?

The relevant animal personality literature has been incorporated throughout our revised manuscript, including in the introduction [79-87] and discussion [224-234] sections.

- I. 167. The mechanisms maintaining relative consistency in behaviour are indeed numerous, and there is great literature on this in the animal personality field. The suggestions outlined here though, “genetic, physiological, and/or temperamental”, are all aspects of the same, i.e. intrinsic factors that regulate the outcome behaviour is individual’s dispositional behavioural tendency. Here, a more appropriate discussion would outline the relative effects of internal and external effects and the potential interactions thereof. Things like maternal effects, developmental effects, the effects that shape the outcome behaviour in a short term (such as facilitation), and the effects of the learning environment are all known to influence personality. Of course, oxytocinergic system is probably at play here, but the outcome behaviour is a result of internal and external effects combined. This could be elaborated with using the recent studies on such effects on personality in other species.

Thank you for this valuable point. We now acknowledge the possibility of both internal and external effects in our discussion, noting that genetic, physiological, developmental, ecological, maternal, and social factors may all be at play [187-195].

- I. 147: The wording is strange here. Yes it is ‘constant’ but behavioural consistency refers to relative stability and rank-order stability. The fact that consolatory tendency declines with age but individuals are relatively stable (based on Fig 2) suggests that there is indeed rather high rank-order stability in your data.

We agree that it is important to clarify what we mean by stable individual variation. As mentioned above, in our revised introduction, we now highlight that we mean rank-order (or relative) stability, rather than absolute stability. Throughout the rest of the paper, we are careful to refer to the relative stability of individual differences—which included editing our wording here accordingly [196-198].

- I. 190: Were infants the targets of aggression frequently enough to allow this inference?

In our dataset, infants were the target of aggression on 334 occasions, a much larger dataset than any previous study investigating infants (which typically involves a few hundred cases for all age categories).

All references above are already cited in manuscript.

References

- Fraser, O. N., Koski, S. E., & Wittig, R. M. (2009). Why are bystanders friendly to recipients of aggression? *Communicative & Integrative Biology* 2(3), 285-291.
- Lessells CM, Boag PT. 1987. Unrepeatable repeatabilities: a common mistake. *AUK* 104:116–121.
- McGraw KO, Wong SP. 1996. Forming inferences about some intraclass correlation coefficients. *Psychol Methods* 1:30–46.
- McAdams, D. P., & Olson, B. D. (2010). Personality Development: Continuity and Change Over the Life Course. *Annual Review of Psychology*, 61(1), 517–542.
- King, J. E., Weiss, A., & Sisco, M. M. (2008). Aping humans: Age and sex effects in

chimpanzee (*Pan troglodytes*) and human (*Homo sapiens*) personality. *Journal of Comparative Psychology*, 122(4), 418–427.

- Ben-Ami Bartal, I., Shan, H., Molasky, N. M. R., Murray, T. M., Williams, J. Z., Decety, J., & Mason, P. (2016). Anxiolytic Treatment Impairs Helping Behavior in Rats. *Frontiers in Psychology*, 7(185), 19–14.

Reviewer #1 (Remarks to the Author):

Review of NCOMMS-16-22607A: Empathic Personalities: Chimpanzees Consistently Vary in the Tendency to Console Others

This revised manuscript concerns examining consolation behavior as a personality trait in chimpanzees and the developmental trajectory and stability of that trait. My previous comments focused on the need to take into account human and other chimpanzee findings related to empathy and similar constructs, the contrast between questionnaire-based and behavior-based measure, and statistical concerns, including how the repeatability was estimated and whether it might be appropriate to correct for multiple tests.

I very much appreciate the efforts the authors made to address my concerns. In particular, their raising the likely possibility that consolation was a manifestation of extraversion and agreeableness helps to better place these findings in with the broader literature. Their noting that studies using both types of measures would help researchers reach stronger conclusions is precisely correct in my view (see, e.g., Weiss & Adams, 2013). A nice way to think about these measures is that they can complement one another. I also appreciate their better explaining their various decisions concerning the analyses. Finally, from what I could see, they dealt with my minor suggestions or at least explained why they decided to take a different path, which is fine. The paper is much better as a result. That said, I have some remaining concerns, which I suspect the authors can easily address. I describe these points below.

The authors note the evidence that empathy increases in humans, but note that "...studies on age differences in empathy have yielded inconsistent findings (references omitted)." I would omit this statement for, although there is definitely evidence for rank-order stability in Agreeableness and similar traits (Roberts & DelVecchio, 2000), the evidence for mean-level increases in Agreeableness is overwhelming. This is clear from longitudinal studies of adults and older individuals, such as that by Terracciano et al. cited in the paper, but also in large studies that cover more than just younger and older adults, and indeed, most of the age range (e.g., McCrae et al., 2002; Soto, John, Gosling, & Potter, 2011; Srivastava, John, Gosling, & Potter, 2003). A comprehensive meta-analysis of the broader literature reached the same conclusion (Roberts, Walton, & Viechtbauer, 2006). The authors did cite papers to support their view in response to my comments.

The references cited by the authors do not strongly support their statement about inconsistent findings. Gröhn et al. highlight the fact that they did not find declines in their longitudinal data. They noted that this finding indicates that these declines probably reflect cohort differences and not developmental trajectories (see Costa & McCrae, 1982 for a discussion of this point). The Helson et al. findings of age-related declines in scores on the CPI empathy (-.014 in Table 2) need to be considered in light of the fact that it does not measure Agreeableness, and, in fact, the CPI itself is content deficient when it comes to Agreeableness (McCrae, Costa, & Piedmont, 1993), and thus is a poor measure of it and similar constructs. The possible exception, incidentally, is the CPI's masculinity-femininity scale (McCrae, et al., 1993), but Helson et al. found that it increased with age (.036 in Table 2), and so their findings are consistent with Agreeableness increases. Finally, the paper by Richter and Kunzmann suggests that cognitive but not emotional aspects of empathy are degraded in older age, which presents some mixed evidence, though their study does not rule out the possibility that their cognitive findings reflect general cognitive declines and not declines specific to empathy.

Second, in referring to the match or mismatch between different kinds of empathy measures, I am happy to let the authors go with this. However, it would be good for them to also cite, even if as a throwaway "see, however..." papers that show associations between Agreeableness and behaviors related to empathy, such as Yarkoni, Ashar, and Wager (2015), and possibly relevant papers cited therein.

Third, in reporting your decision to not correct for the familywise error rate or false discovery rate, I would lay out your logic more clearly. Also, as it is a contentious area, it is probably worth citing the opposing view, too.

Finally, the authors asked me for my views on the recent paper by Almeling et al. Among Barbary macaques, the available data suggests that traits related to Extraversion and Agreeableness combine to form a single personality dimension (Friendliness) in that species (Adams et al., 2015; Konečná, Weiss, Lhota, & Wallner, 2012), which is also true of some other macaque species (see the Adams et al. reference presented earlier). As such, it is possible, but without a study that includes both types of measures, i.e., studies that rely on strong inference (Platt, 1964), I would hesitate to draw strong conclusions either way.

In addition to these comments, I had some minor suggestions concerning style.

l 17: I would say "over" or "more than" instead of ">".

l 23: The phrase "shed light" is a cliché.

l 47: In this context, I think "including" is a better choice than "such as".

l 60: I would omit the word "finally".

l 65: I would delete the "this" at the end of the line.

l 82: I would delete the word "interestingly".

l 118: A reference is needed at the end of the sentence.

Table 1: Please see my earlier comment on reporting p-values.

l 176: I think you need to report that, although there was significant stability, it was weak or modest in size.

l 206: A reference is needed at the end of the sentence.

l 220: Please name the species here (common name should be fine). This will help readers who are unfamiliar with the difference between chimpanzees and monkeys.

l 235: I would omit "here".

l 369: A reference is needed after "chimpanzees".

Thank you for asking me to review this interesting paper. Once again, I hope the authors find my comments helpful.

Alexander Weiss [I sign my reviews]

References

- Adams, M. J., Majolo, B., Ostner, J., Schuelke, O., De Marco, A., Thierry, B., . . . Weiss, A. (2015). Personality structure and social style in macaques. *Journal of Personality and Social Psychology*, 109, 338-353. doi: 10.1037/pspp0000041
- Costa, P. T., Jr., & McCrae, R. R. (1982). An approach to the attribution of aging, period, and cohort effects. *Psychological Bulletin*, 92, 238-250. doi: 10.1037/0033-2909.92.1.238
- Konečná, M., Weiss, A., Lhota, S., & Wallner, B. (2012). Personality in Barbary macaques (*Macaca sylvanus*): Temporal stability and social rank. *Journal of Research in Personality*, 46, 581-590. doi: 10.1016/j.jrp.2012.06.004
- McCrae, R. R., Costa, P. T., Jr., & Piedmont, R. L. (1993). Folk concepts, natural language, and psychological constructs: The California Psychological Inventory and the Five-Factor Model. *Journal of Personality*, 61, 1-26. doi: 10.1111/j.1467-6494.1993.tb00276.x
- McCrae, R. R., Costa, P. T., Jr., Terracciano, A., Parker, W. D., Mills, C. J., De Fruyt, F., & Mervielde, I. (2002). Personality trait development from age 12 to age 18: Longitudinal, cross-sectional and cross-cultural analyses. *Journal of Personality and Social Psychology*, 83, 1456-1468. doi: 10.1037/0022-3514.83.6.1456
- Platt, J. R. (1964). Strong inference. *Science*, 146, 347-353.
- Roberts, B. W., & DelVecchio, W. F. (2000). The rank-order consistency of personality traits from

childhood to old age: A quantitative review of longitudinal studies. *Psychological Bulletin*, 126, 3-25.

Roberts, B. W., Walton, K. E., & Viechtbauer, W. (2006). Patterns of mean-level change in personality traits across the life course: A meta-analysis of longitudinal studies. *Psychological Bulletin*, 132, 1-25. doi: 10.1037/0033-2909.132.1.1

Soto, C. J., John, O. P., Gosling, S. D., & Potter, J. (2011). Age differences in personality traits from 10 to 65: Big Five domains and facets in a large cross-sectional sample. *Journal of Personality and Social Psychology*, 100, 330-348. doi: 10.1037/a0021717

Srivastava, S., John, O. P., Gosling, S. D., & Potter, J. (2003). Development of personality in early and middle adulthood: Set like plaster or persistent change? *Journal of Personality and Social Psychology*, 84, 1041-1053. doi: 10.1037/0022-3514.84.5.1041

Weiss, A., & Adams, M. J. (2013). Differential behavioral ecology. In C. Carere & D. Maestriperi (Eds.), *Animal personalities: Behavior, physiology and evolution* (pp. 96-123). Chicago, IL: University of Chicago Press.

Yarkoni, T., Ashar, Y. K., & Wager, T. D. (2015). Interactions between donor Agreeableness and recipient characteristics in predicting charitable donation and positive social evaluation. *PeerJ*, 3, e1089. doi: 10.7717/peerj.1089

Reviewer #2 (Remarks to the Author):

The authors report consistent inter-individual differences in (post-conflict) affiliation in a group of chimpanzees. As such it is an interesting and well conducted study. However, these findings are not very novel and consequently not of great interest to the wider community. The authors, however, insist in calling what they find 'Empathic personality' which, if true, may indeed be more newsworthy. Whereas the behavioural results the authors report are suggestive of empathic processes, the authors nevertheless report no results on proximate mechanisms that back that claim. Moreover, I still have serious concerns about the behavioural definitions of what is measured:

First, the authors score post-conflict affiliation, yet do not score base-line affiliation as a control. Which is surprising as the PC/MC method is the standard set for looking at 'consolation' by the head of this lab. Consequently, it is unclear what we are looking at: Consistent inter-individual differences in post-conflict affiliation, or 'just' consistent inter-individual differences in affiliation.

Second, I'm still not convinced that this (post-conflict) affiliation really constitutes consolation. Whereas the authors now do mention that there are alternative explanations for post-conflict affiliation, they immediately, in my opinion falsely, dismiss these alternatives. They back this up by previous studies, which are, however, almost all from their own lab. Moreover, they mention that their definition excludes the alternatives as it only concerns affiliation towards the victim and not to the aggressor. Yet one of the alternative explanations is that this affiliation prevents redirected aggression by the victim towards the affiliating individual.

Finally, the authors consider consolation in chimpanzees as a proxy for empathy in chimpanzees, yet provide no evidence that it is. All the studies they mention only provide an indirect link. Of course, one can speculate about the proximate mechanism, and I'm willing to accept empathy as one of the possibilities. But it must be clear that this is speculation rather than fact.

Minor comments:

I. 51. I'm not aware of such an extensive literature on animals' capability to understand the emotions of others. Please provide references.

I. 58. Whereas it may follow predictions derived from an empathy-based hypothesis, this doesn't make it a fact yet. It may also follow predictions of other hypotheses.

l. 63consolation behavior is considered BY THE AUTHORS homologous.....

l. 87 it is not a measure of empathy. If anything, only indirect.

l. 169 again this is considered so by the authors and is not a general belief, nor a fact. Rather say something like 'that it is suggested to be'

l. 238 I'm surprised that the authors refer here to the Tennie et al. 2016 study. This study reports a lack of prosocial preferences among chimpanzees, and contrary to what the authors state here, does not suggest that empathic abilities facilitate other-oriented processes and behaviour. In fact, that paper does not mention the word empathy at all.

Reviewer #3 (Remarks to the Author):

Thank you for your thorough responses to my earlier concerns, and the corresponding revisions to the manuscript. Specifically, the methods are now solid and allow also direct comparisons to other personality studies, the result section's clarity has much improved, and you have incorporated the relevant literature on human and chimpanzee personality. As such, I am happy to recommend the revision for publication.

I spotted an editorial error in the reference list, which is entirely my bad actually. The ref. I asked you to add, Fraser et al 2009, was missing the last author's name (copy paste error of sorts?), and it now lacks it in your ref. list too. Sorry about that!

Best, Sonja Koski (reviewer 3)

Dear Dr Webb,

Your manuscript entitled "Empathic Personalities: Chimpanzees Consistently Vary in the Tendency to Console Others" has been seen again by three referees, whose comments are appended below. You will see from their comments copied below that, while Reviewers 1 and 3 are largely satisfied with the revision, Reviewer 2 continues to raise quite substantial concerns that must be addressed. In light of these comments, we cannot accept the manuscript for publication, but would be interested in considering a revised version that addresses these concerns.

In order to consider this paper further, we would expect a revision to include data controlling for baseline levels of affiliation tendency (as described by Reviewer 2). Other editors and I feel that these data are essential for bolstering the conclusions and ruling out alternative interpretations. We further request the text regarding evidence of empathy be rephrased throughout the paper to instead refer to consolation and affiliation.

If the revision process takes significantly longer than three months, we will be happy to reconsider your paper at a later date, as long as nothing similar has been accepted for publication at Nature Communications or published elsewhere in the meantime.

We are committed to providing a fair and constructive peer-review process. Do not hesitate to contact us if you wish to discuss the revision or if there are specific requests from the reviewers that you believe are technically impossible or unlikely to yield a meaningful outcome.

When resubmitting your paper, we also ask that you ensure that your manuscript complies with our editorial policies. Specifically, please ensure that the following requirements are met, and any relevant checklists are completed and uploaded as a Related Manuscript file type with the revised article:

Reporting requirements for life sciences research: http://www.nature.com/article-assets/npg/ncomms/authors/ncomms_lifesciences_checklist.pdf

Please use the following link to submit your revised manuscript, point-by-point response to the referees' comments (which should be in a separate document to any cover letter) and any completed checklist:

<http://mts-ncomms.nature.com/cgi-bin/main.plex?el=A1S3oNI6B4GZfC7I6A9ftd45gNJpP5YLHQ8OZbjT69wZ>

Please do not hesitate to contact me if you have any questions or would like to discuss the required revisions further. Thank you for the opportunity to review your work.

Best regards,
Alexa

Alexa McKay, PhD
Associate Editor
Nature Communications

Reviewers' comments:

Reviewer #1 (Remarks to the Author):

Review of NCOMMS-16-22607A: Empathic Personalities: Chimpanzees Consistently Vary in the Tendency to Console Others

This revised manuscript concerns examining consolation behavior as a personality trait in chimpanzees and the developmental trajectory and stability of that trait. My previous comments focused on the need to take into account human and other chimpanzee findings related to empathy and similar constructs, the contrast between questionnaire-based and behavior-based measure, and statistical concerns, including how the repeatability was estimated and whether it might be appropriate to correct for multiple tests.

I very much appreciate the efforts the authors made to address my concerns. In particular, their raising the likely possibility that consolation was a manifestation of extraversion and agreeableness helps to better place these findings in with the broader literature. Their noting that studies using both types of measures would help researchers reach stronger conclusions is precisely correct in my view (see, e.g., Weiss & Adams, 2013). A nice way to think about these measures is that they can complement one another. I also appreciate their better explaining their various decisions concerning the analyses. Finally, from what I could see, they dealt with my minor suggestions or at least explained why they decided to take a different path, which is fine. The paper is much better as a result. That said, I have some remaining concerns, which I suspect the authors can easily address. I describe these points below.

The authors note the evidence that empathy increases in humans, but note that "...studies on age differences in empathy have yielded inconsistent findings (references omitted)." I would omit this statement for, although there is definitely evidence for rank-order stability in Agreeableness and similar traits (Roberts & DelVecchio, 2000), the evidence for mean-level increases in Agreeableness is overwhelming. This is clear from longitudinal studies of adults and older individuals, such as that by Terracciano et al. cited in the paper, but also in large studies that cover more than just younger and older adults, and indeed, most of the age range (e.g., McCrae et al., 2002; Soto, John, Gosling, & Potter, 2011; Srivastava, John, Gosling, & Potter, 2003). A comprehensive meta-analysis of the broader literature reached the same conclusion (Roberts, Walton, & Viechtbauer, 2006). The authors did cite papers to support their view in response to my comments. The references cited by the authors do not strongly support their statement about inconsistent findings. Grühn et al. highlight the fact that they did not find declines in their longitudinal data. They noted that this finding indicates that these declines probably reflect cohort differences and not developmental trajectories (see Costa & McCrae, 1982 for a discussion of this point). The Helson et al. findings of age-related declines in scores on the CPI empathy (-.014 in Table 2) need to be considered in light of the fact that it does not measure Agreeableness, and, in fact, the CPI itself is content

deficient when it comes to Agreeableness (McCrae, Costa, & Piedmont, 1993), and thus is a poor measure of it and similar constructs. The possible exception, incidentally, is the CPI's masculinity-femininity scale (McCrae, et al., 1993), but Helson et al. found that it increased with age (.036 in Table 2), and so their findings are consistent with Agreeableness increases. Finally, the paper by Richter and Kunzmann suggests that cognitive but not emotional aspects of empathy are degraded in older age, which presents some mixed evidence, though their study does not rule out the possibility that their cognitive findings reflect general cognitive declines and not declines specific to empathy.

We have now removed this statement to avoid any potential confusion [34-39], but would respectfully submit that directly comparing or equating findings from the Agreeableness literature with those in the empathy literature can be tricky (depending on which facets are emphasized—e.g., the empathy literature often focuses more on self-regulatory capacities like negative emotion control). However, the point is not central to our paper and we will therefore defer to the reviewer's expertise in this area.

Second, in referring to the match or mismatch between different kinds of empathy measures, I am happy to let the authors go with this. However, it would be good for them to also cite, even if as a throwaway "see, however..." papers that show associations between Agreeableness and behaviors related to empathy, such as Yarkoni, Ashar, and Wager (2015), and possibly relevant papers cited therein.

Thanks for this suggestion; we agree that it is a point worth making [209].

Third, in reporting your decision to not correct for the familywise error rate or false discovery rate, I would lay out your logic more clearly. Also, as it is a contentious area, it is probably worth citing the opposing view, too.

We appreciate this comment, but are tight on space and references, and thus do not think this paper is the appropriate forum to delve into these issues any further.

Finally, the authors asked me for my views on the recent paper by Almeling et al. Among Barbary macaques, the available data suggests that traits related to Extraversion and Agreeableness combine to form a single personality dimension (Friendliness) in that species (Adams et al., 2015; Konečná, Weiss, Lhota, & Wallner, 2012), which is also true of some other macaque species (see the Adams et al. reference presented earlier). As such, it is possible, but without a study that includes both types of measures, i.e., studies that rely on strong inference (Platt, 1964), I would hesitate to draw strong conclusions either way.

Thank you for this response and for providing us with some additional references of interest.

In addition to these comments, I had some minor suggestions concerning style.

I 17: I would say "over" or "more than" instead of ">".

Agreed [16].

I 23: The phrase “shed light” is a cliché.

We have now replaced this phrase [23].

I 47: In this context, I think “including” is a better choice than “such as”.

Agreed [30].

I 60: I would omit the word “finally”.

Omitted [61].

I 65: I would delete the “this” at the end of the line.

Deleted [66].

I 82: I would delete the word “interestingly”.

Deleted [83].

I 118: A reference is needed at the end of the sentence.

We have now provided supporting references [122-123].

Table 1: Please see my earlier comment on reporting p-values.

We have followed this helpful suggestion in the main body of the paper, but in the Table make an exception in the service of formatting (columns appear less organized when the values are rounded to different numbers of decimal places).

I 176: I think you need to report that, although there was significant stability, it was weak or modest in size.

We have now done so [173 and 181].

I 206: A reference is needed at the end of the sentence.

We do not include a reference here because the statement refers to the general approaches taken within each field rather than a specific study or subset of studies.

I 220: Please name the species here (common name should be fine). This will help readers who are unfamiliar with the difference between chimpanzees and monkeys.

Good suggestion—thanks [226].

I 235: I would omit “here”.

Omitted [241].

I 369: A reference is needed after “chimpanzees”.

We have now added a reference [378].

Thank you for asking me to review this interesting paper. Once again, I hope the authors find my comments helpful.

Your comments have been very insightful; thanks once again for your helpful feedback!

Alexander Weiss [I sign my reviews]

References

- Adams, M. J., Majolo, B., Ostner, J., Schuelke, O., De Marco, A., Thierry, B., . . . Weiss, A. (2015). Personality structure and social style in macaques. *Journal of Personality and Social Psychology*, 109, 338-353. doi: 10.1037/pspp0000041
- Costa, P. T., Jr., & McCrae, R. R. (1982). An approach to the attribution of aging, period, and cohort effects. *Psychological Bulletin*, 92, 238-250. doi: 10.1037/0033-2909.92.1.238
- Konečná, M., Weiss, A., Lhota, S., & Wallner, B. (2012). Personality in Barbary macaques (*Macaca sylvanus*): Temporal stability and social rank. *Journal of Research in Personality*, 46, 581-590. doi: 10.1016/j.jrp.2012.06.004
- McCrae, R. R., Costa, P. T., Jr., & Piedmont, R. L. (1993). Folk concepts, natural language, and psychological constructs: The California Psychological Inventory and the Five-Factor Model. *Journal of Personality*, 61, 1-26. doi: 10.1111/j.1467-6494.1993.tb00276.x
- McCrae, R. R., Costa, P. T., Jr., Terracciano, A., Parker, W. D., Mills, C. J., De Fruyt, F., & Mervielde, I. (2002). Personality trait development from age 12 to age 18: Longitudinal, cross-sectional and cross-cultural analyses. *Journal of Personality and Social Psychology*, 83, 1456-1468. doi: 10.1037/0022-3514.83.6.1456
- Platt, J. R. (1964). Strong inference. *Science*, 146, 347-353.
- Roberts, B. W., & DelVecchio, W. F. (2000). The rank-order consistency of personality traits from childhood to old age: A quantitative review of longitudinal studies. *Psychological Bulletin*, 126, 3-25.
- Roberts, B. W., Walton, K. E., & Viechtbauer, W. (2006). Patterns of mean-level change in personality traits across the life course: A meta-analysis of longitudinal studies. *Psychological Bulletin*, 132, 1-25. doi: 10.1037/0033-2909.132.1.1
- Soto, C. J., John, O. P., Gosling, S. D., & Potter, J. (2011). Age differences in personality traits from 10 to 65: Big Five domains and facets in a large cross-sectional sample. *Journal of Personality and Social Psychology*, 100, 330-348. doi: 10.1037/a0021717
- Srivastava, S., John, O. P., Gosling, S. D., & Potter, J. (2003). Development of personality in early and middle adulthood: Set like plaster or persistent change? *Journal of Personality and Social Psychology*, 84, 1041-1053. doi: 10.1037/0022-3514.84.5.1041
- Weiss, A., & Adams, M. J. (2013). Differential behavioral ecology. In C. Carere & D. Maestriperi (Eds.), *Animal personalities: Behavior, physiology and evolution* (pp. 96-123). Chicago, IL: University of Chicago Press.
- Yarkoni, T., Ashar, Y. K., & Wager, T. D. (2015). Interactions between donor Agreeableness and recipient characteristics in predicting charitable donation and positive social evaluation. *PeerJ*, 3, e1089. doi: 10.7717/peerj.1089

Reviewer #2 (Remarks to the Author):

The authors report consistent inter-individual differences in (post-conflict) affiliation in a group of chimpanzees. As such it is an interesting and well conducted study. However, these findings are not very novel and consequently not of great interest to the wider community. The authors, however, insist in calling what they find 'Empathic personality' which, if true, may indeed be more newsworthy. Whereas the behavioural results the authors report are suggestive of empathic processes, the authors nevertheless report no results on proximate mechanisms that back that claim. Moreover, I still have serious concerns about the behavioural definitions of what is measured:

Please find our detailed responses to your comments below.

First, the authors score post-conflict affiliation, yet do not score base-line affiliation as a control. Which is surprising as the PC/MC method is the standard set for looking at 'consolation' by the head of this lab. Consequently, it is unclear what we are looking at: Consistent inter-individual differences in post-conflict affiliation, or 'just' consistent inter-individual differences in affiliation.

Although previous studies have already established consolation as a phenomenon that does not merely reflect a general affiliative tendency, we do not want to convey the message that we disregarded this important dimension. Thus, in the revision, we have included an additional analysis that controls for bystander baseline affiliation [332-338]. This analysis demonstrates that individual differences in bystander post-conflict affiliation remain significant even after accounting for a general tendency to affiliate with others [116-119; 180-181]. Thank you for pointing out the need to include such an analysis, as it has served to improve the integrity of our findings.

Second, I'm still not convinced that this (post-conflict) affiliation really constitutes consolation. Whereas the authors now do mention that there are alternative explanations for post-conflict affiliation, they immediately, in my opinion falsely, dismiss these alternatives. They back this up by previous studies, which are, however, almost all from their own lab. Moreover, they mention that their definition excludes the alternatives as it only concerns affiliation towards the victim and not to the aggressor. Yet one of the alternative explanations is that this affiliation prevents redirected aggression by the victim towards the affiliating individual.

Although we are willing to acknowledge the existence of alternative accounts, the consolation explanation is the most applicable to our observations (for reasons which were outlined in our previous response). Nonetheless, we have edited aspects of our introduction so as not to sound immediately dismissive of these alternatives [57-60]. From our previous reply, we will reemphasize here that detailed functional analyses on consolation behavior in Great Apes, from multiple labs, have failed to find support for the self-protection hypothesis mentioned by Reviewer 2 (e.g., Wittig & Boesch, 2010; Romero et al., 2010; Romero & de Waal, 2010; Clay & de Waal, 2013; Palagi & Norscia, 2013)—with only one exception (Koski & Sterck, 2009).

Finally, the authors consider consolation in chimpanzees as a proxy for empathy in chimpanzees, yet provide no evidence that it is. All the studies they mention only provide an indirect link. Of course, one can speculate about the proximate mechanism, and I'm

willing to accept empathy as one of the possibilities. But it must be clear that this is speculation rather than fact.

In the revision, we are careful to emphasize that we found individual differences in consolation rather than individual differences in empathy, reflected in our new title and corresponding changes to the abstract and main body of the paper (detailed below under minor comments, see lines [1, 15, 57-59, 79, 105, 172, 244]).

Nonetheless, although we appreciate these concerns, we do not consider the connection between consolation and empathy to be speculative, especially after the many behavioral and recent neuroscientific studies—but we are willing to point to this evidence and explain that there may be alternative accounts. However, we do not know what these alternative accounts might be, and unfortunately Reviewer 2 does not point us in a direction that is helpful in this regard.

Research not only coming from our lab, but again the work of many other labs, has supported the link between consolation and empathy. Notably, findings from independent studies in Great Apes have shown that: a) the main function of consolation is to reduce the victim's distress, b) alternative functional hypotheses have been rejected (see above), and c) the behavior fits other predictions derived from the empathy-based theory (Kutsukake & Castles, 2004; Palagi et al., 2004; Palagi et al., 2006; Fraser et al., 2008; Romero et al., 2010; Romero & de Waal, 2010; Clay & de Waal, 2013a, 2013b; Palagi & Norscia, 2013, see Fraser & Bugnyar, 2010; Palagi et al., 2014; Burkett et al., 2016, for similar results in other species). As such, this link is generally not considered to be tenuous. Our usage of the term empathy was accepted by Reviewers 1 and 3, and is consistent with several upcoming reviews on mammalian empathy—e.g., see: de Waal, F. B. M., & Preston, S. D. (in press). Mammalian empathy: Behavioral manifestations and neural basis. *Nature Reviews: Neuroscience*.

Minor comments:

I. 51. I'm not aware of such an extensive literature on animals' capability to understand the emotions of others. Please provide references.

We provide examples of this literature starting in the subsequent sentence and throughout the rest of this paragraph [49-64]. The last several decades has witnessed increased interest in animals' emotions as well as their reactions to other's emotions. These studies have focused on a wide array of species (from chimpanzees to rodents to domestic species) and research areas (from behavior to neuroscience). Just to give a few examples: Parr, 2001; Berntson et al., 1989; Watanabe & Ono, 1986, Langford et al., 2006, 2010; Plotnik & de Waal, 2014; Phelps & LeDoux, 2005; Mendl, et al., 2010, de Waal, 2011; Panksepp & Watt, 2011; Nagasawa et al., 2011; Schmidt & Cohn, 2001; Ghazanfar & Logothetis, 2003; Izumi & Kojima, 2004; Faragó et al., 2010; Racca et al., 2012; Müller et al., 2015; Custance et al., 2012; Andics et al., 2014. Due to length constraints, in the main text we limited the references to those more directly related with consolation.

I. 58. Whereas it may follow predictions derived from a empathy-based hypothesis, this doesn't make it a fact yet. It may also follow predictions of other hypotheses.

We also do not believe that the term ‘hypothesis’ makes it a fact. Nonetheless, we have been more careful in our use of the term empathy [e.g., 1, 15, 79, 105, 172, 244] and included further consideration of alternative hypotheses [57-59].

I. 63consolation behavior is considered BY THE AUTHORS homologous.....

We have added the word ‘often’ to qualify this statement [64-65]. Although consolation is considered functionally similar to human empathy-related responding not just by the authors but also by many independent groups who study these phenomena (see above), we have revised our language throughout to describe that we found individual differences in consolation rather than empathy per se, and to describe consolation’s ‘reported’ or ‘suggested’ or ‘putative’ links to empathy rather than a direct relation.

I. 87 it is not a measure of empathy. If anything, only indirect.

We have now removed the term empathy from this sentence [89].

I. 169 again this is considered so by the authors and is not a general belief, nor a fact. Rather say something like 'that it is suggested to be'

We have taken the reviewer’s suggestion here as well [172].

I. 238 I'm surprised that the authors refer here to the Tennie et al. 2016 study. This study reports a lack of prosocial preferences among chimpanzees, and contrary to what the authors state here, does not suggest that empathic abilities facilitate other-oriented processes and behaviour. In fact, that paper does not mention the word empathy at all.

We apologize for any confusion. Here we meant to refer to recent general interest in the topic, which should include current debates and opposing viewpoints. However, the reviewer is correct in noting that this sentence was potentially misleading in this way, and thus we have edited it to better reflect our intended message [243-246].

Reviewer #3 (Remarks to the Author):

Thank you for your thorough responses to my earlier concerns, and the corresponding revisions to the manuscript. Specifically, the methods are now solid and allow also direct comparisons to other personality studies, the result section's clarity has much improved, and you have incorporated the relevant literature on human and chimpanzee personality. As such, I am happy to recommend the revision for publication.

I spotted an editorial error in the reference list, which is entirely my bad actually. The ref. I asked you to add, Fraser et al 2009, was missing the last author's name (copy paste error of sorts?), and it now lacks it in your ref. list too. Sorry about that!
Best, Sonja Koski (reviewer 3)

Thank you for your encouraging feedback and helpful suggestions. We have now corrected the Fraser et al. (2009) reference.

Reviewer #1 (Remarks to the Author):

Review of NCOMMS-16-22607B: Consolatory Personalities: Chimpanzees Consistently Vary in the Tendency to Console Others

This is the second version of a manuscript that describes individual differences in consolatory behavior in chimpanzees. I took the time to read this manuscript, the recent reviews by myself and the other reviewers, and the authors' responses to those reviews.

My main concern with the previous version of the manuscript was that the authors needed to show more nuance with regards to how they dealt with some of the human literature, and the existing literature on nonhuman primates. In response, the authors opted to omit this text and noted in their letter that making direct comparisons across the literature with different methods is not straightforward. I think they appropriately addressed this comment, and would agree with their statement. My remaining comments were minor and, in my view, addressed adequately, either in the covering letter or in the revised manuscript.

I had a few minor comments, which can probably be addressed in page proofs. I don't need to see another draft.

1. I did not see any reason for the authors to change the title of the paper. It was better before.
2. For the text spanning lines 42 to 43, I would delete the parentheses.
3. In line 174, nonhuman is spelled with a hyphen. It probably should not be.

Thank you for asking me to review this manuscript. I hope the authors find these remaining comments helpful.

Alexander Weiss [I sign my reviews]

REVIEWERS' COMMENTS:

Reviewer #1 (Remarks to the Author):

Review of NCOMMS-16-22607B: Consolatory Personalities: Chimpanzees Consistently Vary in the Tendency to Console Others

This is the second version of a manuscript that describes individual differences in consolatory behavior in chimpanzees. I took the time to read this manuscript, the recent reviews by myself and the other reviewers, and the authors' responses to those reviews.

My main concern with the previous version of the manuscript was that the authors needed to show more nuance with regards to how they dealt with some of the human literature, and the existing literature on nonhuman primates. In response, the authors opted to omit this text and noted in their letter that making direct comparisons across the literature with different methods is not straightforward. I think they appropriately addressed this comment, and would agree with their statement. My remaining comments were minor and, in my view, addressed adequately, either in the covering letter or in the revised manuscript.

I had a few minor comments, which can probably be addressed in page proofs. I don't need to see another draft.

1. I did not see any reason for the authors to change the title of the paper. It was better before.

At the Editor's discretion, we will revert to a version of our former title: "Individual variation in chimpanzee consolation reflects empathic personalities" (must remove punctuation to adhere to journal formatting requirements).

2. For the text spanning lines 42 to 43, I would delete the parentheses.

Deleted.

3. In line 174, nonhuman is spelled with a hyphen. It probably should not be.

We checked and it looks like the term is hyphenated in this journal. Nonetheless, we were inconsistent in our previous formatting—so thank you for pointing this out.

Thank you for asking me to review this manuscript. I hope the authors find these remaining comments helpful.

We very much appreciate your thoughtful and helpful feedback at all stages of the revision process, it has greatly improved our manuscript!

Alexander Weiss [I sign my reviews]